# SteerConf: Steering LLMs for Confidence Elicitation

**Ziang Zhou**
Department of Computing
The Hong Kong Polytechnic University
Hong Kong, China
fduscottzhou@outlook.com

**Tianyuan Jin**
Department of Electrical and Computer Engineering
National University of Singapore
Singapore
tianyuan@nus.edu.sg

**Jieming Shi**[*]
Department of Computing
The Hong Kong Polytechnic University
Hong Kong, China
jieming.shi@polyu.edu.hk

**Qing Li**
Department of Computing
The Hong Kong Polytechnic University
Hong Kong, China
qing-prof.li@polyu.edu.hk

## Abstract

Large Language Models (LLMs) exhibit impressive performance across diverse domains but often suffer from overconfidence, limiting their reliability in critical applications. We propose SteerConf, a novel framework that systematically steers LLMs' confidence scores to improve their calibration and reliability. SteerConf introduces three key components: (1) a steering prompt strategy that guides LLMs to produce confidence scores in specified directions (e.g., conservative or optimistic) by leveraging prompts with varying steering levels; (2) a steered confidence consistency measure that quantifies alignment across multiple steered confidences to enhance calibration; and (3) a steered confidence calibration method that aggregates confidence scores using consistency measures and applies linear quantization for answer selection. SteerConf operates without additional training or fine-tuning, making it broadly applicable to existing LLMs. Experiments on seven benchmarks spanning professional knowledge, common sense, ethics, and reasoning tasks, using advanced LLM models (GPT-3.5, LLaMA 3, GPT-4), demonstrate that SteerConf significantly outperforms existing methods, often by a significant margin. Our findings highlight the potential of steering the confidence of LLMs to enhance their reliability for safer deployment in real-world applications. The implementation is at https://github.com/scottjiao/SteerConf.

## 1 Introduction

Large Language Models (LLMs) are widely used in applications such as AI-driven chatbots [29, 21, 13], medical diagnosis [9, 30, 24], code generation [23, 15], legal document analysis [40, 19], and education [34, 14]. Ensuring the reliability and trustworthiness of LLMs is critical, particularly in high-stakes domains like healthcare, legal analysis, and autonomous systems. However, LLMs often exhibit *overconfidence*, producing predictions that fail to align with their true likelihood of correctness, even when uncertain [38], posing significant challenges for their practical deployment.

To obtain prediction confidence from LLMs, *confidence elicitation* methods treat LLMs as *black boxes*, using prompts to elicit self-assessed *verbalized confidence* scores [38, 31]. Unlike traditional approaches [22, 25] that rely on white-box access to internal model information, some LLMs are closed-source, with only commercial APIs, such as GPT-3.5 [28] and GPT-4 [29], available for use.

---

[*]Corresponding author.

39th Conference on Neural Information Processing Systems (NeurIPS 2025).

These APIs do not provide access to internal details like token likelihoods, making traditional confidence elicitation infeasible. Additionally, fine-tuning LLMs for confidence elicitation is prohibitively expensive, limiting its practicality for most researchers.

Recent studies focus on eliciting confidence from LLMs via prompts in a black-box manner to improve confidence calibration, aligning confidence with accuracy and enhancing failure prediction—measuring the ability of LLMs to assign high confidence to correct predictions and low confidence to incorrect ones [38]. Specifically, [38] propose a systematic framework that probes LLMs with various prompts, samples multiple responses to elicit confidence scores, and aggregates them to obtain a final confidence score. For instance, a Top-K prompting strategy [31] asks an LLM to provide the top-K most likely answers and their associated confidence scores for a question. A sample prompt is: "Provide your K best guesses and the confidence that each is correct (0% to 100%) for the following question."

**Observation.** However, the prompting strategies explored so far, do not *explictly steer the direction* of confidence in a controlled manner, i.e., whether the LLMs should be more conservative or optimistic in their confidence scores. Importantly, we find that if we explicitly instruct LLMs to be more conservative or optimistic, the LLMs can be steered to produce confidence scores in a specific direction. For example, we can steer LLMs to be *very cautious* in answering questions and producing confidence scores, compared with the vanilla prompt, as shown in the example below.

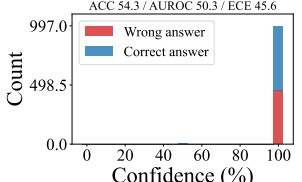 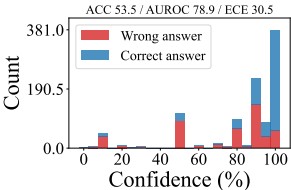

(a) Vanilla Verbalized Confidence  (b) Very-Cautious Steered Prompt

Figure 1: Vanilla and *very cautious* Prompt Steered Confidence on Object Counting data with GPT-3.5.

- *Vanilla Prompt:* "Read the question carefully, analyze step by step, and provide your answer along with your confidence. Confidence reflects how likely you believe your answer is correct."

- *Very Cautious Prompt:* The Vanilla Prompt with an additional instruction: "You should adopt a very cautious approach and assign low confidence to most of your answers."

Figure 1 compares the effects of vanilla and very cautious steered prompts on the Object Counting Dataset [33] using GPT-3.5 [28]. In Figure 1(a), the vanilla prompt results in severe overconfidence, with LLMs assigning 100% confidence to answers that are only 50% accurate for counting tasks. In contrast, Figure 1(b) shows that the *very cautious* steered prompt yields better-calibrated confidence scores, where confidence levels more closely align with actual accuracy. For example, answers with 100% confidence achieve approximately 90% accuracy, while those with 10% confidence correspond to about 10% accuracy. Quantitatively, the very cautious prompt improves calibration as measured by ECE and failure prediction as measured by AUROC, achieving a 29% increase in Area Under the Receiver Operating Characteristic Curve (AUROC, higher is better) and a 16% reduction in Expected Calibration Error (ECE, lower is better). These findings demonstrate the effectiveness of steering prompts in enhancing the reliability of LLM confidence scores.

**Our Contributions**. We propose SteerConf, a novel framework that leverages the steerability of LLMs' confidence scores to enhance their reliability. SteerConf comprises three key components: (1) We propose a steering prompt strategy to guide LLMs in generating confidence scores in a specified direction (e.g., conservative or optimistic) by employing prompts with varying steering intensities. For each question, we sample multiple responses, capturing both answers and their associated confidence scores across different steering levels. (2) We introduce a steered confidence consistency measure to evaluate the alignment of confidence scores across multiple steered responses, leveraging this consistency to enhance confidence calibration. (3) We design a steered confidence calibration method that leverages the consistency measures to aggregate confidence scores and applies linear quantization to identify the most reliable answer based on the calibrated confidence. SteerConf is a simple yet effective framework that can be directly applied to existing LLMs without requiring additional training or fine-tuning.

Comprehensive experiments across seven benchmarks, covering professional knowledge, common sense, ethics, and reasoning tasks, using three state-of-the-art models (GPT-3.5, LLaMA 3, GPT-4), demonstrate that SteerConf significantly outperforms vanilla verbalized confidence and existing

calibration methods. For example, SteerConf achieves up to a 39.8% reduction in ECE on GSM8K dataset with GPT-3.5 and a 43.3% improvement in AUROC on the Object Counting dataset with GPT-4 in the CoT setting.

## 2 Preliminaries

### 2.1 Task Statement

**Verbalized Confidence.** Confidence represents a model's certainty in its predictions, typically expressed as a probability (e.g., "80% confidence"). A score closer to 1 indicates high certainty, while a score near 0 reflects uncertainty. However, for large language models (LLMs), internal confidence scores are not directly accessible due to their black-box nature. Instead, confidence is elicited through verbalized responses obtained via prompting, where the model explicitly states its confidence in natural language (e.g., "I am 80% confident in answer A") [38, 31].

**Verbalized Confidence Elicitation.** This process involves modifying the original query prompt to explicitly request confidence scores from LLMs. For example, given the question "Which one is bigger, 9.8 or 9.10? Provide your answer," the prompt can be rephrased as "Which one is bigger, 9.8 or 9.10? Provide your answer along with your confidence (%)." The model might then respond with "9.8, 80%" instead of just "9.8," thereby verbalizing its confidence level as 80%.

**Confidence Calibration and Failure Prediction.** Verbalized confidence scores elicited from LLMs are often miscalibrated, meaning the reported confidence does not accurately reflect the true likelihood of correctness. For instance, a model might state "80% confidence" for answers that are correct only 50% of the time. *Confidence calibration* addresses this issue by aligning the model's confidence scores with the actual accuracy of predictions [11]. Ideally, predictions with 80% confidence should exhibit 80% accuracy. This is crucial for applications where confidence scores guide decision-making, such as in medical diagnosis or autonomous driving. The effectiveness of confidence calibration is commonly evaluated using metrics such as ECE. *Failure prediction* focuses on identifying incorrect predictions by leveraging confidence scores. It is formulated as a binary classification task, where correct predictions are labeled as 0 and incorrect ones as 1. The performance of failure prediction is measured using AUROC, which assesses the model's ability to rank incorrect predictions below correct ones [1]. Detailed computations and evaluations are provided in the experiments.

### 2.2 Related Work

**Elicitation of Confidence in LLMs.** The concept of verbalized confidence, where models directly express confidence through natural language, was introduced in [22], but their work primarily focuses on fine-tuned models trained on specific datasets. Zero-shot verbalized confidence in LLMs remains unexplored in their study. Similarly, [25] trains external models to estimate confidence based on internal model representations, which are inaccessible in closed-source LLMs. While [44] examines the impact of confidence expressions, it does not provide explicit confidence scores to users. [31] explores prompting strategies, proposing the Top-K verbalized prompting method. The unified framework summarized in [38], which considers sample consistency, is the most closely related to our approach. In contrast, our work introduces a novel framework to steer LLM confidence in a desired direction, addressing gaps left by these prior studies.

**Confidence Calibration.** Modern AI systems often exhibit unreliable confidence estimates, frequently displaying overconfidence in their predictions [11, 26, 37]. Calibration techniques address this issue by aligning confidence scores with actual accuracy rates [11, 26]. Two primary approaches have been explored: (1) scaling methods, which adjust confidence scores to improve reliability [11, 4, 43], and (2) binning methods, which group predictions by confidence levels to assess and correct miscalibration [42, 43]. For LLMs, studies have revealed significant calibration challenges. For instance, [16] demonstrated that models like T5 and GPT-2 are poorly calibrated in question-answering tasks. Similarly, [2] found that while standard language models face calibration issues, pretraining can enhance confidence reliability. Conversely, [17] observed that larger LLMs, when properly prompted, exhibit improved calibration in multiple-choice tasks. However, a critical limitation of these studies is their reliance on access to internal model computations (e.g., logits), which is infeasible for closed-source systems like GPT-4. This limitation underscores the need for alternative confidence

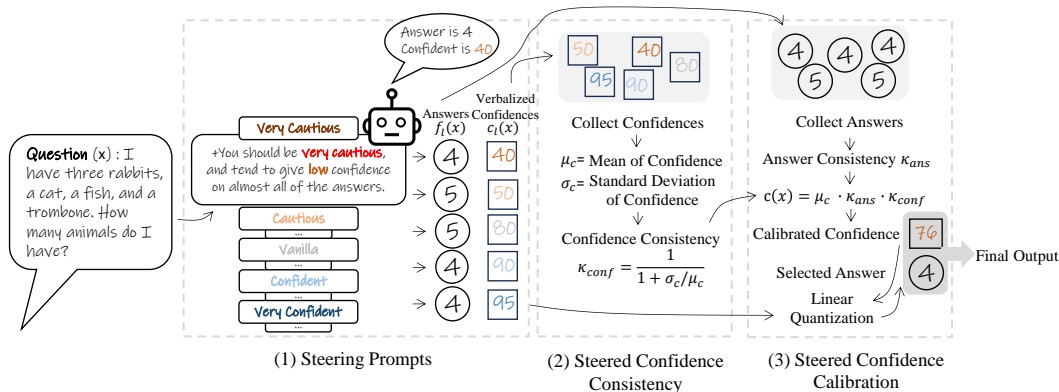

Figure 2: Overview of the SteerConf framework, comprising three components: Steering Prompting, Steered Confidence Consistency, and Steered Confidence Calibration. Five steering levels, from very cautious to very confident, elicit verbalized confidences and answers from the LLM. Confidence consistency $\kappa_{conf}$ evaluates the stability of these confidences. The calibrated confidence $c(x)$ is computed by combining the mean confidence $\mu_c$, confidence consistency $\kappa_{conf}$, and answer consistency $\kappa_{ans}$, and the final answer $f(x)$ is selected based on $c(x)$.

estimation methods that operate without requiring internal model access or modifications, motivating our proposed framework.

# 3 SteerConf: Steering the Confidence of LLMs

Figure 2 illustrates the overall framework of SteerConf, comprising three key components: Steering Prompting (Section 3.1), Steered Confidence Consistency (Section 3.2), and Steered Confidence Calibration (Section 3.3).

Given an input question $x$, we first construct a set of steering prompts $P_{conf}^l$ with varying steering levels (e.g., very cautious) to elicit steered answers $f_l(x)$ and confidence scores $c_l(x)$ from the LLM. Next, the Steered Confidence Consistency module computes the confidence consistency $\kappa_{conf}$ by analyzing the mean $\mu_c$ and standard deviation $\sigma_c$ of the sampled confidence scores $c_l(x)$. The calibrated confidence score $c(x)$ is then derived by combining $\mu_c$, $\kappa_{conf}$, and the answer consistency $\kappa_{ans}$, which is determined by the relative frequency of the most frequent answer among $f_l(x)$. Finally, the calibrated confidence score $c(x)$ is used to select the final answer $f(x)$ via linear quantization, mapping $c(x)$ to the corresponding steering index in the confidence distribution.

## 3.1 Steering Prompts

As discussed in Section 1, while LLMs often exhibit overconfidence, explicitly prompting them to be cautious can result in lower confidence scores. To leverage this behavior, we propose a *steering prompting strategy* to systematically elicit and sample confidence scores from LLMs, enabling more accurate confidence estimation. Specifically, this strategy employs *steering prompts* that explicitly instruct the LLM to adopt varying levels of caution or confidence.

Specifically, we employ a set of steering levels to guide the LLM's confidence elicitation from very cautious to very confident: {*very cautious*, *cautious*, *vanilla*, *confident*, *very confident*}. The steering level "very cautious" in the prompt instructs the LLM to be very cautious, with which the LLM tends to be more conservative in its confidence estimation. In the steering prompt, we also include the original question to be answered, and the LLM is asked to provide both the answer and the confidence score. A brief example of very-cautious prompt is shown in the introduction. In the steering prompt, the steering level can be replaced with different levels, and the LLM is asked to provide the answer and confidence score for each steering level. The detailed prompt design is shown in Appendix B.

Formally, given an input text $x$, we can apply $2\ell + 1$ steering levels via steering prompts to the LLM, generating $2\ell + 1$ prediction-confidence pairs. Let $P_{conf}^l$ be the $l$-th steering prompt, where

$l \in \{-\ell, -\ell+1, \ldots, 0, \ldots, \ell-1, \ell\}$. Negative steering levels $-\ell, -\ell+1, \ldots, -1$ represent cautious steering levels, while positive steering levels $1, 2, \ldots, \ell$ represent confident steering levels. $-\ell$ and $\ell$ represent the most cautious and most confident steering levels, respectively, while $0$ represents the vanilla steering level. In other words, $P_{\text{conf}}^{-\ell}$ is the most cautious steering prompt, $P_{\text{conf}}^{\ell}$ is the most confident steering prompt, and $P_{\text{conf}}^{0}$ is the vanilla steering prompt, which is the same with original verbalized confidence elicitation process.

In our setting, we set $\ell = 2$, which means we have five steering levels: {*very cautious*, *cautious*, *vanilla*, *confident*, *very confident*}, a moderate granularity, which is sufficient to demonstrate the effectiveness of our method. Our framework is flexible and can be extended to more steering levels by increasing $\ell$. Given a question $x$, we construct the steering prompt set $\{P_{\text{conf}}^{l}\}_{l=-\ell}^{\ell}$, and for each steering prompt $P_{\text{conf}}^{l}$, we can elicit the answer and confidence score from the LLM to get the prediction-confidence pair $(f_l(x), c_l(x))$, where $f_l(x)$ is the answer and $c_l(x)$ is the confidence score. The confidence score $c_l(x)$ is a real number between 0 and 1, representing the LLM's confidence in its answer $f_l(x)$. By applying different steering levels, we can probe the LLM's confidence in different directions and intensities, which can help us to better understand the LLM's confidence and uncertainty, to facilitate the accurate confidence estimation.

## 3.2 Steered Confidence Consistency

Given a set of sampled steered answers $\{f_l(x)\}_{l=-\ell}^{\ell}$ and their confidence scores $\{c_l(x)\}_{l=-\ell}^{\ell}$, we aim to compute a calibrated confidence score $c(x)$ that reflects the model's certainty about answering the input question $x$. In previous studies [38, 6, 5, 20], it has been shown that the consistency of multiple responses for a given question from LLMs can provide meaningful guidance on model confidence. For example, the degree of agreement among multiple answers has been used to serve as a measure of confidence for LLMs [38], which however overlooks the consistency of the verbalized confidences returned by LLMs, which may contain rich information of model's interior certainty.

Therefore, in this section, we propose a *steered confidence consistency* measure that considers the consistency of the steered verbalized confidence scores $\{c_l(x)\}_{l=-\ell}^{\ell}$ to estimate the model's certainty about answering the input question $x$. The intuition is that, even under different steering levels, if the LLM is able to respond with consistent confidence scores, it indicates that the model is confident about its answer, even with explicit prompt interventions. The higher the consistency of steered confidence scores, the more reliable the LLM is about its answer regarding the input question. To quantify this, we first compute the mean confidence $\mu_c$ of the steered confidences $\{c_l(x)\}_{l=-\ell}^{\ell}$ as in Eq. (1), representing the average certainty level, and the standard deviation $\sigma_c$ in Eq. (2), measuring how consistently this certainty is maintained across steering levels from $-\ell$ to $\ell$. Intuitively, higher $\mu_c$ with lower $\sigma_c$ indicates stronger confidence consistency.

Intuitively, the ratio $\frac{\sigma_c}{\mu_c}$, known as the coefficient of variation, naturally normalizes the dispersion of a distribution relative to the mean, showing the extent of variability in relation to the mean of the observations. A smaller value of $\frac{\sigma_c}{\mu_c}$ indicates that the data points tend to be very close to the mean and are more consitent, while a larger value indicates that the data points are spread out over a wider range of values, being less consistent. Therefore, we propose the confidence consistency $\kappa_{\text{conf}} = \frac{1}{1+\sigma_c/\mu_c}$ in Eq. (3), which is a bounded metric between 0 and 1, where $\kappa_{\text{conf}} = 1$ indicates perfect consistency with $\sigma_c = 0$ and $\kappa_{\text{conf}} = 0$ indicates no consistency with infinite $\sigma_c$.

$$\mu_c = \frac{1}{2\ell+1} \sum_{l=-\ell}^{\ell} c_l(x) \tag{1}$$

$$\sigma_c = \sqrt{\frac{1}{2\ell+1} \sum_{l=-\ell}^{\ell} (c_l(x) - \mu_c)^2} \tag{2}$$

$$\kappa_{\text{conf}} = \frac{1}{1+\sigma_c/\mu_c} \tag{3}$$

## 3.3 Steered Confidence Calibration

For a question $x$, we aim to compute the calibrated confidence score $c(x)$ that reflects the model's certainty about answering the question $x$. In a nutshell, the calibrated confidence score $c(x)$ should be high when the model is confident about its answer, and low when the model is uncertain about its answer. To achieve this, we propose to consider the mean confidence $\mu_c$, confidence consistency $\kappa_{\text{conf}}$, and answer consistency $\kappa_{\text{ans}}$ that is computed from the steered answers $\{f_l(x)\}_{l=-\ell}^{\ell}$.

First, the answer consistency $\kappa_{ans}$ is computed based on the relative frequency of the answers [38]. For each answer $y \in \{f_l(x)\}_{l=-\ell}^{\ell}$, its frequency $\text{freq}(y)$ is defined as the proportion of times $y$ occurs among all steered answers: $\text{freq}(y) = \frac{1}{2\ell+1} \sum_{l=-\ell}^{\ell} \mathbb{I}(f_l(x) = y)$, where $\mathbb{I}(\cdot)$ is the indicator function. The answer consistency is then defined as the maximum relative frequency of the most frequent answer: $\kappa_{ans} = \max_{y \in \{f_l(x)\}_{l=-\ell}^{\ell}} \text{freq}(y)$. This reflects the degree of agreement among the answers, with higher $\kappa_{ans}$ indicating greater consistency and reliability of the model's predictions.

Then, the calibrated confidence $c(x)$ is defined as the product of the average steered confidence $\mu_c$, answer consistency $\kappa_{\text{ans}}$, and confidence consistency $\kappa_{\text{conf}}$ as in Eq. (4). The factor $\mu_c$ considers all the steered confidence scores, while $\kappa_{\text{ans}}$ and $\kappa_{\text{conf}}$, the two consistency metrics, are used to adjust the average steered confidence $\mu_c$ based on LLMs' self-consistency in terms of answers and confidence scores. If the LLM shows low consistency in either answers or confidence scores, the final calibrated confidence $c(x)$ will be downweighted accordingly, and vise versa. This design ensures the robustness of the final calibrated confidence $c(x)$ against isolated high-confidence outliers that may arise from the black-box nature of LLMs, as any single inconsistency component ($\kappa_{\text{ans}}$ or $\kappa_{\text{conf}}$) can sufficiently downweight potentially miscalibrated $\mu_c$.

$$c(x) = \mu_c \cdot \kappa_{\text{ans}} \cdot \kappa_{\text{conf}} \tag{4}$$

After obtaining the final calibrated confidence $c(x)$ from Eq. (4), the next step is to select the final answer $f(x)$ from the set of steered answers $\{f_l(x)\}_{l=-\ell}^{\ell}$. The selection should align with the calibrated confidence $c(x)$, which represents the model's certainty.

Since $c(x)$ may not directly match any of the steered confidence scores $\{c_l(x)\}_{l=-\ell}^{\ell}$, we employ a mapping mechanism to determine the most appropriate answer. We map $c(x)$ to the steering index space using linear quantization. First, we compute the range of observed steered confidences:

$$c_{\min} = \min_l c_l(x), \quad c_{\max} = \max_l c_l(x). \tag{5}$$

Next, we map $c(x)$ to a discrete steering index $j$ using:

$$j = \left\lfloor \frac{c(x) - c_{\min}}{c_{\max} - c_{\min}} \cdot (2\ell + 1) \right\rfloor, \tag{6}$$

where $\lfloor \cdot \rfloor$ denotes the floor operation, ensuring $j$ is an integer.

Finally, the selected answer $f(x)$ is determined as:

$$f(x) = \begin{cases} f_{-\ell}(x), & \text{if } j < -\ell, \\ f_j(x), & \text{if } -\ell \leq j \leq \ell. \end{cases}$$

This approach ensures that the answer $f(x)$ is consistent with the calibrated confidence $c(x)$, while maintaining numerical stability through explicit range normalization.

# 4 Experiments

## 4.1 Setup

**Datasets.** We assess confidence estimation quality across five categories of reasoning tasks: (1) *Commonsense Reasoning* using Sports Understanding dataset (Sport) [18] and StrategyQA (StrategyQA) [7] from BigBench [8]; (2) *Arithmetic Reasoning* evaluated on GSM8K (GSM8K) [3]; (3) *Symbolic Reasoning* covering Date Understanding (DateUnd) [36] and Object Counting (ObjCnt) [33]; (4) *Professional Knowledge* tested through Law (Law) from MMLU [12]; and (5) *Ethical Knowledge* examined via Business Ethics (Ethics) in MMLU [12].

**Table 1**

| Model | GSM8K | DateUnd | ObjCnt | StrategyQA | Sport | Law | Ethics | Avg |
|---|---|---|---|---|---|---|---|---|
| LLaMA3 | 74.8 | 37.8 | 50.2 | 26.6 | 30.6 | 30.9 | 15.0 | 38.0 |
| +SteerConf | 45.2 | 11.8 | 34.6 | 29.3 | 16.4 | 8.5 | 26.5 | **24.6** |
| GPT-3.5 | 62.6 | 60.2 | 45.6 | 29.6 | 25.3 | 43.2 | 26.0 | 41.8 |
| +SteerConf | 22.8 | 33.0 | 27.5 | 14.9 | 12.2 | 24.0 | 10.8 | **20.7** |
| GPT-4 | 53.5 | 25.7 | 23.7 | 16.8 | 18.7 | 19.4 | 5.1 | 23.3 |
| +SteerConf | 27.5 | 19.5 | 18.0 | 13.5 | 13.6 | 11.2 | 9.9 | **16.2** |

(a) ECE ↓ (Lower is Better)

| Model | GSM8K | DateUnd | ObjCnt | StrategyQA | Sport | Law | Ethics | Avg |
|---|---|---|---|---|---|---|---|---|
| LLaMA3 | 53.7 | 50.3 | 50.3 | 58.8 | 51.5 | 51.0 | 42.3 | 51.1 |
| +SteerConf | 67.1 | 61.0 | 67.2 | 59.0 | 53.8 | 58.9 | 62.9 | **61.4** |
| GPT-3.5 | 55.8 | 56.6 | 50.3 | 53.3 | 52.8 | 51.7 | 54.8 | 53.6 |
| +SteerConf | 82.9 | 60.5 | 82.5 | 58.6 | 61.5 | 60.6 | 71.0 | **68.2** |
| GPT-4 | 52.0 | 50.5 | 50.4 | 55.6 | 57.9 | 56.6 | 84.1 | 58.2 |
| +SteerConf | 83.9 | 64.2 | 72.7 | 63.7 | 63.3 | 59.7 | 77.6 | **69.3** |

(b) AUROC ↑ (Higher is Better)

| Model | GSM8K | DateUnd | ObjCnt | StrategyQA | Sport | Law | Ethics | Avg |
|---|---|---|---|---|---|---|---|---|
| LLaMA3 | 81.7 | 38.2 | 50.2 | 33.0 | 34.8 | 40.8 | 30.3 | 44.1 |
| +SteerConf | 88.5 | 51.7 | 66.1 | 32.1 | 37.7 | 47.0 | 52.3 | **53.6** |
| GPT-3.5 | 76.9 | 66.0 | 46.1 | 37.6 | 32.3 | 54.8 | 37.5 | 50.2 |
| +SteerConf | 93.5 | 64.1 | 76.4 | 43.5 | 41.7 | 64.4 | 51.7 | **62.2** |
| GPT-4 | 55.3 | 26.5 | 24.3 | 30.7 | 23.9 | 41.3 | 58.4 | 37.2 |
| +SteerConf | 83.5 | 35.9 | 47.3 | 37.5 | 33.2 | 46.2 | 45.0 | **46.9** |

(c) PR-N ↑ (Higher is Better)

| Model | GSM8K | DateUnd | ObjCnt | StrategyQA | Sport | Law | Ethics | Avg |
|---|---|---|---|---|---|---|---|---|
| LLaMA3 | 21.3 | 62.5 | 49.9 | 77.2 | 66.9 | 61.0 | 71.5 | 58.6 |
| +SteerConf | 28.1 | 68.0 | 62.3 | 79.6 | 67.1 | 66.2 | 79.1 | **64.3** |
| GPT-3.5 | 30.0 | 41.5 | 54.5 | 66.2 | 70.8 | 47.5 | 69.6 | 54.3 |
| +SteerConf | 60.4 | 54.4 | 83.3 | 71.1 | 76.1 | 52.3 | 80.2 | **68.3** |
| GPT-4 | 47.3 | 74.5 | 76.5 | 77.6 | 82.6 | 67.5 | 94.5 | 74.4 |
| +SteerConf | 76.8 | 81.2 | 84.4 | 81.4 | 84.4 | 71.3 | 92.0 | **81.6** |

(d) PR-P ↑ (Higher is Better)

Table 1: Comparing SteerConf with vanilla verbalized confidence elicitation of LLMs. ECE > 0.25, AUROC, PR-P, PR-N < 0.6 denote significant deviation from ideal performance. The best is in bold.

**Table 2**

| Model | GSM8K | DateUnd | ObjCnt | StrategyQA | Sport | Law | Ethics | Avg |
|---|---|---|---|---|---|---|---|---|
| LLaMA3(CoT) | 5.0 | 13.7 | 8.7 | 11.8 | 7.7 | 22.8 | 7.8 | 11.1 |
| +SteerConf | 7.8 | 1.0 | 6.5 | 8.6 | 5.4 | 20.9 | 5.0 | **7.9** |
| GPT-3.5(CoT) | 20.3 | 30.8 | 41.8 | 26.0 | 20.5 | 44.3 | 24.8 | 29.8 |
| +SteerConf | 6.5 | 6.7 | 19.7 | 14.5 | 10.8 | 16.0 | 15.1 | **12.7** |
| GPT-4(CoT) | 6.5 | 6.6 | 4.9 | 18.5 | 9.2 | 23.0 | 6.1 | 10.7 |
| +SteerConf | 4.3 | 8.4 | 1.6 | 11.5 | 5.6 | 9.9 | 9.3 | **7.2** |

(a) ECE ↓ (Lower is Better)

| Model | GSM8K | DateUnd | ObjCnt | StrategyQA | Sport | Law | Ethics | Avg |
|---|---|---|---|---|---|---|---|---|
| LLaMA3(CoT) | 55.1 | 54.3 | 50.0 | 64.6 | 74.1 | 54.3 | 54.2 | 58.1 |
| +SteerConf | 81.2 | 70.7 | 87.6 | 74.5 | 79.4 | 64.7 | 81.2 | **77.0** |
| GPT-3.5(CoT) | 56.2 | 49.8 | 50.1 | 56.4 | 62.7 | 53.0 | 65.2 | 56.2 |
| +SteerConf | 85.6 | 76.0 | 82.5 | 67.5 | 66.4 | 61.5 | 86.7 | **75.2** |
| GPT-4(CoT) | 52.1 | 75.1 | 50.0 | 68.8 | 65.0 | 59.5 | 87.6 | 65.5 |
| +SteerConf | 86.0 | 81.5 | 93.3 | 70.3 | 75.2 | 68.4 | 93.0 | **81.1** |

(b) AUROC ↑ (Higher is Better)

| Model | GSM8K | DateUnd | ObjCnt | StrategyQA | Sport | Law | Ethics | Avg |
|---|---|---|---|---|---|---|---|---|
| LLaMA3(CoT) | 12.4 | 13.4 | 8.7 | 29.7 | 38.3 | 38.2 | 26.1 | 23.8 |
| +SteerConf | 41.3 | 24.0 | 49.0 | 43.2 | 47.5 | 50.6 | 51.7 | **43.9** |
| GPT-3.5(CoT) | 26.5 | 28.0 | 42.0 | 36.8 | 46.9 | 55.5 | 45.7 | 40.2 |
| +SteerConf | 70.9 | 49.7 | 74.7 | 48.9 | 52.2 | 60.9 | 75.0 | **61.8** |
| GPT-4(CoT) | 10.5 | 47.6 | 4.9 | 48.7 | 27.7 | 46.2 | 83.2 | 38.4 |
| +SteerConf | 59.3 | 61.8 | 44.9 | 38.9 | 48.7 | 55.4 | 74.9 | **54.9** |

(c) PR-N ↑ (Higher is Better)

| Model | GSM8K | DateUnd | ObjCnt | StrategyQA | Sport | Law | Ethics | Avg |
|---|---|---|---|---|---|---|---|---|
| LLaMA(CoT) | 95.4 | 89.3 | 91.3 | 85.2 | 89.2 | 66.5 | 83.0 | 85.7 |
| +SteerConf | 97.8 | 92.6 | 98.1 | 90.0 | 92.8 | 73.6 | 93.1 | **91.1** |
| GPT-3.5(CoT) | 80.5 | 71.5 | 58.2 | 71.5 | 70.2 | 48.4 | 75.8 | 68.0 |
| +SteerConf | 92.1 | 88.4 | 83.2 | 77.7 | 75.0 | 57.3 | 91.8 | **80.8** |
| GPT-4(CoT) | 93.7 | 94.2 | 95.1 | 81.4 | 88.0 | 68.3 | 92.8 | 87.6 |
| +SteerConf | 97.3 | 94.0 | 99.2 | 87.6 | 91.1 | 75.6 | 97.9 | **91.8** |

(d) PR-P ↑ (Higher is Better)

Table 2: Comparing SteerConf with verbalized confidence elicitation of LLMs with CoT. ECE > 0.25, AUROC, PR-P, PR-N < 0.6 denote significant deviation from ideal performance. The best is in bold.

**Tasks and Metrics.** We evaluate our method SteerConf on two tasks: confidence calibration and failure prediction, following the protocol in [38]. For confidence calibration, we report Expected Calibration Error (ECE), which measures the average absolute difference between predicted confidence and empirical accuracy; lower ECE indicates better calibration. For failure prediction, we use the Area Under the Receiver Operating Characteristic curve (AUROC), where higher values (closer to 1.0) indicate better discrimination between correct and incorrect predictions. To account for class imbalance due to varying accuracy, we further report AUPRC-Positive (PR-P) and AUPRC-Negative (PR-N), which respectively measure the model's ability to prioritize incorrect and correct predictions in a precision-recall framework. Formal definitions of all metrics are provided in Appendix A.

**LLM Models.** We evaluate several widely-used LLMs: GPT-3.5 [28], LLaMA3 [10], and GPT-4 [29]. LLaMA3 is used in its 70B parameter version with 4-bit quantization. Note that experiments with GPT-4 incurred a cost of approximately 1500 USD due to its higher pricing. We first compare these LLMs using vanilla verbalized confidence elicitation, and then evaluate each LLM with CoT prompting [35] for confidence elicitation. Results are reported in Table 1 and Table 2 in Section 4.2. Details on the application of CoT are provided in Appendix B.

**Baselines.** We also compare against sampling-based and prompting-based baselines summarized in [38], including Misleading [38], Self-Random [38], and Top-K [31], as reported in Table 3. All baselines use an LLM with CoT prompting as the backbone. Self-Random [38] leverages the model's inherent randomness by inputting the same prompt multiple times, and consistency aggregation metric [38] is applied. Misleading [38] evaluates the model's uncertainty by introducing misleading

| Method | GSM8K | | Law | | DateUnd | | StrategyQA | | Ethics | | Avg | |
|---|---|---|---|---|---|---|---|---|---|---|---|---|
| | ECE | AUROC | ECE | AUROC | ECE | AUROC | ECE | AUROC | ECE | AUROC | ECE | AUROC |
| GPT-3.5(CoT) | 10.1 | 54.8 | 39.7 | 52.2 | 23.4 | 57.4 | 22.0 | 59.8 | 30.0 | 56.0 | 25.0 | 56.4 |
| +Top-K | 19.6 | 58.5 | 16.7 | 58.9 | 26.1 | 74.2 | 14.0 | 61.3 | 12.4 | 73.3 | 17.8 | 65.2 |
| +Misleading | 8.03 | 88.6 | 18.3 | 59.3 | 20.5 | 67.3 | 21.8 | 61.5 | 17.8 | 71.3 | 17.3 | 69.6 |
| +Self-Random | 6.28 | 92.7 | 26.0 | 65.6 | 17.0 | 66.8 | 23.3 | 60.8 | 20.7 | 79.0 | 18.7 | 73.0 |
| +SteerConf (ours) | 6.5 | 85.6 | 16.0 | 61.5 | 6.7 | 76.0 | 14.5 | 67.5 | 15.1 | 86.7 | **11.7** | **75.4** |
| LLaMA3(CoT) | 5.0 | 55.1 | 22.8 | 54.3 | 13.7 | 54.3 | 11.8 | 64.6 | 7.8 | 54.2 | 12.2 | 56.5 |
| +Top-K | 10.2 | 61.1 | 23.0 | 49.7 | 27.5 | 52.2 | 21.1 | 55.0 | 11.1 | 52.4 | 18.6 | 54.1 |
| +Misleading | 4.4 | 83.8 | 18.9 | 59.7 | 18.4 | 67.8 | 11.7 | 65.3 | 14.0 | 75.0 | 13.5 | 70.3 |
| +Self-Random | 2.2 | 79.3 | 27.1 | 64.6 | 7.2 | 66.8 | 17.3 | 60.1 | 15.9 | 55.3 | 13.9 | 65.2 |
| +SteerConf (ours) | 6.1 | 81.2 | 3.3 | 64.7 | 10.0 | 70.7 | 4.3 | 74.5 | 13.6 | 81.2 | **7.5** | **74.5** |

Table 3: Comparison with baselines on GPT-3.5 and LLaMA3 with CoT. The best is in bold and the second best is underlined.

| Method | GSM8K | | Law | | DateUnd | | StrategyQA | | Ethics | | Avg | |
|---|---|---|---|---|---|---|---|---|---|---|---|---|
| | ECE | AUROC | ECE | AUROC | ECE | AUROC | ECE | AUROC | ECE | AUROC | ECE | AUROC |
| SteerConf w/o Steering Prompting | 2.3 | 80.0 | 13.2 | 66.7 | 10.0 | 70.2 | 6.1 | 72.5 | 10.2 | 59.0 | 8.4 | 69.7 |
| Self-Random + Consistency | 2.2 | 79.3 | 27.1 | 64.6 | 7.2 | 66.8 | 17.3 | 60.1 | 15.9 | 55.3 | 13.9 | 65.2 |
| SteerConf + Consistency | 2.4 | 71.7 | 25.9 | 63.8 | 7.5 | 68.2 | 17.2 | 61.4 | 14.3 | 68.5 | 13.5 | 66.7 |
| Self-Random + Avg-Conf | 2.3 | 79.3 | 27.1 | 64.4 | 8.8 | 68.3 | 17.2 | 59.9 | 16.3 | 65.0 | 14.3 | 65.4 |
| SteerConf + Avg-Conf | 2.8 | 71.3 | 25.8 | 63.3 | 8.6 | 67.1 | 17.4 | 61.3 | 16.7 | 73.5 | 14.3 | 67.3 |
| SteerConf (ours) | 6.1 | 81.2 | 3.3 | 64.7 | 10.0 | 70.7 | 4.3 | 74.5 | 13.6 | 81.2 | **7.5** | **74.5** |

Table 4: Component ablations and alternative aggregation schemes on LLaMA3 with CoT. Lower ECE values indicate better calibration, while higher AUROC values reflect improved performance.

information and observing whether the model maintains its initial answer, inspired by human behavior under confidence and consistency aggregation metric [38] is applied. Top-K [31] prompts the model to generate $K$ answer-confidence pairs in a single response, pair-rank metric [38] is applied. For Misleading and Self-Random, we use $M = 5$ samples; for Top-K, we set $K = 5$ answer-confidence pairs. We adopt GPT-3.5 and LLaMA3 both with CoT.

## 4.2 Effectiveness Results

**Comparison with LLMs with and without CoT.** Table 1 and Table 2 report the performance of our method SteerConf versus vanilla verbalized confidence elicitation on LLaMA3, GPT-3.5, and GPT-4, without and with CoT prompting, respectively, across ECE, AUROC, PR-N, and PR-P metrics (all in %). The last column in each table shows the average across datasets. Our method consistently outperforms vanilla verbalized confidence elicitation across nearly all metrics and datasets, both with and without CoT, demonstrating improved confidence calibration and failure detection. For instance, in Table 1(a), GPT-3.5 with SteerConf achieves an average ECE of 20.7%, substantially better than the vanilla ECE of 41.8%. On ObjCnt, SteerConf attains an AUROC of 82.5%, significantly better than 50.3% for vanilla GPT-3.5 (Table 1(b)). CoT prompting further improves overall performance of all LLMs (Table 2), yet SteerConf maintains a clear advantage. For example, in Table 2(b), GPT-4(CoT) with SteerConf achieves an average AUROC of 81.1%, versus 65.5% for vanilla. On the Ethics dataset, LLaMA3(CoT) with SteerConf yields a PR-N of 51.7%, markedly higher than the vanilla baseline (Table 2(c)).

**Comparison with Other Baselines.** Table 3 presents a comparison of our method SteerConf with the Top-K, Misleading, and Self-Random baselines on LLaMA3 and GPT-3.5 with CoT. Across all datasets and metrics, SteerConf consistently outperforms the baselines, highlighting its superiority in both confidence calibration and failure detection. For instance, on GPT-3.5(CoT), SteerConf achieves an average ECE of 11.7%, substantially lower than the 17.3% of the Misleading baseline. On LLaMA3(CoT), SteerConf attains an average AUROC of 74.5%, outperforming all other baselines.

**Ablation Study.** We perform an ablation study to assess the impact of steering prompting and aggregation components, using LLaMA3 with CoT as the backbone. For the steering prompts component, removing it yields "SteerConf w/o Steering Prompting". For the aggregation component, we replace our proposed aggregation with Consistency and Avg-Conf, denoted as "SteerConf + Consistency" and "SteerConf + Avg-Conf". Ablating both components results in the Self-Random approach with Consistency or Avg-Conf aggregation. The results are reported in Table 4, which

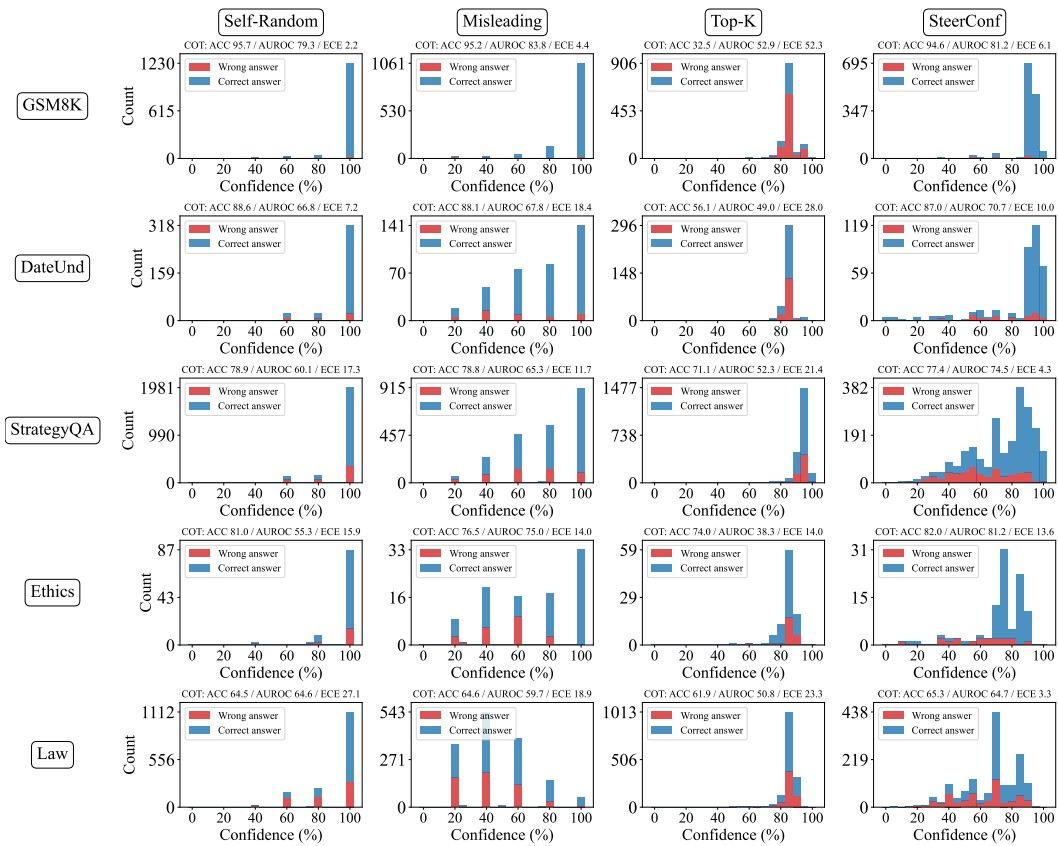

Figure 3: Confidence histograms of SteerConf and baselines for LLaMA3 with CoT. The rows are different datasets, and the columns are different methods.

demonstrates that each component contributes to improved performance, supporting the effectiveness of our method's design.

## 4.3 Empirical Confidence Distributions

Figure 3 presents the confidence histograms of SteerConf and baselines on LLaMA3 with CoT. Each row corresponds to a dataset, and each column shows the confidence distributions for Self-Random, Misleading, Top-K, and our method SteerConf. The x-axis is confidence, and the y-axis is the number of samples. Blue and red bars represent the confidence distributions for correct and incorrect answers, respectively. Compared to the baselines, SteerConf produces confidence distributions that are better separated between correct and incorrect answers, while baselines such as Self-Random tend to yield overconfident or less discriminative distributions. These results further validate the effectiveness of our method in improving confidence calibration, consistent with the quantitative findings in Section 4.2.

## 4.4 Detailed Calibration Distributions

We provide the detailed calibration curves [32] of SteerConf and baselines on LLaMA3 with CoT. The results are presented in Figure 4. As depicted in Figure 4, the confidence distribution from vanilla verbalization is concentrated on a few discrete values. In contrast, SteerConf yields a much more uniform distribution. Such continuous confidence values are more amenable to quantization, thereby improving model reliability. Moreover, we observe a better alignment between the calibrated confidence from SteerConf and the actual accuracy. On the Sport and StrategyQA datasets, for example, SteerConf shows a consistent pattern of alignment across the full spectrum of confidence values, demonstrating the effectiveness of our approach.

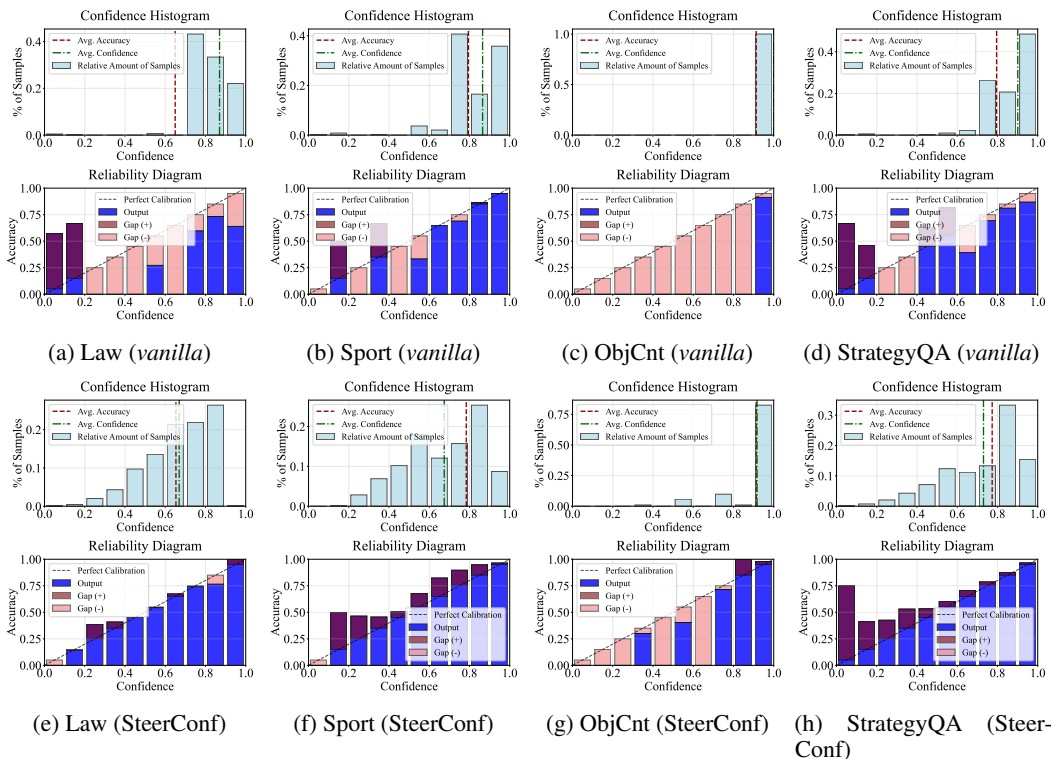

Figure 4: Expected Calibration Error (ECE) plots and reliable diagrams for LLaMA3-70B model. The columns correspond to four tasks: Law, Sport, ObjCnt, and StrategyQA. The top row shows results of vanilla verbalized confidence, while the bottom row shows results from our SteerConf. In each plot, the upper part shows the ECE plots, and the lower part shows the reliable diagrams.

## 5   Conclusion, Limitations, and Future Work

In this paper, we introduced SteerConf, a framework for improving LLM confidence reliability through semantic steering. Our approach addresses the critical issue of overconfidence, which undermines trustworthiness in high-stakes applications. By leveraging carefully designed prompts, we systematically steer LLM confidence in desired directions. The framework comprises three key components: a steering prompt strategy, a steered confidence consistency measure, and a steered confidence calibration method. Together, these components enhance confidence calibration without requiring model internals or fine-tuning. Comprehensive evaluations across seven benchmarks with state-of-the-art LLMs demonstrate consistent improvements over existing methods. This work has significant implications for deploying LLMs in critical domains where reliable confidence estimation is essential. By offering an effective black-box approach, SteerConf enables trustworthy AI without requiring specialized expertise or access to model internals.

**Limitations and Future Work.** It is common to mannually design prompts for LLMs, and our method is no exception. Future work will explore automatic prompt generation to address this. Additionally, using multiple prompts increases computational costs; we aim to develop adaptive strategies to optimize prompt usage based on task and model needs. Following prior work, our current tasks focus on numerical answers (GSM8K) or multiple-choice questions (Law), where ambiguity is minimal and answer consistency is both sufficient and effective. Extending to more open-ended tasks is an important future direction.

**Broader Impact.** This work improves LLM confidence elicitation, enhancing their reliability and trustworthiness in high-stakes applications. While we foresee no direct negative societal impacts, LLM misuse remains a concern. We advocate for responsible AI practices to mitigate such risks.

## Acknowledgments

This work is supported by grants from the Research Grants Council of Hong Kong Special Administrative Region, China (No. PolyU 25201221, No. PolyU 15205224), NSFC No. 62202404, and Smart Cities Research Institute (SCRI) P0051036-P0050643.

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

# A    Task Statement And Metric

**Confidence Calibration.** LLMs often exhibit misaligned confidence scores, frequently overestimating the reliability of their predictions. The goal of confidence calibration is to align these confidence scores with the actual accuracy of the predictions. To evaluate calibration performance, we use the Expected Calibration Error (ECE) [27, 39, 41] as the primary metric.

Given a set of input samples $\{x_i\}_{i=1}^N$ with corresponding labels $\{y_i\}_{i=1}^N$, LLM predictions $\{f(x_i)\}_{i=1}^N$, and confidence scores $\{c(x_i)\}_{i=1}^N$, we partition the samples into $B$ bins based on their confidence scores: $x_i \in B_b$ if $c(x_i) \in \left(\frac{b-1}{B}, \frac{b}{B}\right)$. The Expected Calibration Error (ECE) is computed as:

$$\text{ECE} = \sum_{b=1}^B \frac{|B_b|}{N} \left|\text{acc}(B_b) - \text{conf}(B_b)\right|, \tag{7}$$

where $B_b$ is the set of samples in the $b$-th bin, $|B_b|$ is the number of samples in the $b$-th bin, $\text{acc}(B_b) = \frac{1}{|B_b|} \sum_{x_i \in B_b} \mathbb{I}(f(x_i) = y_i)$ is the accuracy of the samples in the $b$-th bin, and $\text{conf}(B_b) = \frac{1}{|B_b|} \sum_{x_i \in B_b} c(x_i)$ is the average confidence score of the samples in the $b$-th bin. Here, $\mathbb{I}(\cdot)$ is the indicator function, which returns 1 if its argument is true, and 0 otherwise.

**Failure Prediction.** Failure prediction involves determining whether an LLM's prediction is correct based on its confidence score. This task is evaluated using the Area Under the Receiver Operating Characteristic Curve (AUC-ROC) [1]. Given a set of input samples $\{x_i\}_{i=1}^N$, the task is framed as a binary classification problem, where the input is the confidence score $c(x_i)$, and the label indicates the correctness of the LLM's prediction, $\mathbb{I}(f(x_i) = y_i)$. The AUC-ROC is defined as:

$$\text{AUC-ROC} = \int_0^1 \text{TPR}(t)\, d\text{FPR}(t), \tag{8}$$

where $\text{TPR}(t)$ (True Positive Rate) and $\text{FPR}(t)$ (False Positive Rate) at threshold $t$ are computed as:

$$\text{TPR}(t) = \frac{1}{N} \sum_{i=1}^N \mathbb{I}(c(x_i) \geq t)\mathbb{I}(f(x_i) = y_i), \tag{9}$$

$$\text{FPR}(t) = \frac{1}{N} \sum_{i=1}^N \mathbb{I}(c(x_i) \geq t)\left(1 - \mathbb{I}(f(x_i) = y_i)\right). \tag{10}$$

**Precision-Recall Perspective.** To complement AUC-ROC, we evaluate failure prediction using the Area Under the Precision-Recall Curve for the positive class (PR-P) and the negative class (PR-N). These metrics are particularly informative under class imbalance. For PR-P, the positive class corresponds to correct predictions ($f(x_i) = y_i$), while for PR-N, the positive class corresponds to incorrect predictions ($f(x_i) \neq y_i$). The Precision-Recall Curve (PRC) is parameterized by threshold $t$, with precision and recall defined as:

$$\text{Precision}_p(t) = \frac{\sum_{i=1}^N \mathbb{I}(c(x_i) \geq t)\mathbb{I}(f(x_i) = y_i)}{\sum_{i=1}^N \mathbb{I}(c(x_i) \geq t)}, \tag{11}$$

$$\text{Recall}_p(t) = \frac{\sum_{i=1}^N \mathbb{I}(c(x_i) \geq t)\mathbb{I}(f(x_i) = y_i)}{\sum_{i=1}^N \mathbb{I}(f(x_i) = y_i)}, \tag{12}$$

for PR-P, and analogously for PR-N by replacing $\mathbb{I}(f(x_i) = y_i)$ with $\mathbb{I}(f(x_i) \neq y_i)$ in both the numerator and denominator. The AUPRC values are computed as:

$$\text{PR-P} = \int_0^1 \text{Precision}_p(t)\, d\text{Recall}_p(t), \quad \text{PR-N} = \int_0^1 \text{Precision}_n(t)\, d\text{Recall}_n(t). \tag{13}$$

PR-P emphasizes the model's ability to identify correct predictions across confidence thresholds, while PR-N focuses on detecting failures. Together, these metrics provide a nuanced evaluation of calibration quality, particularly under skewed class distributions.

# B    Steering Prompts Used

In this section, we present the detailed prompts used in our experiments for LLMs under both CoT and non-CoT settings. The steering levels {*very cautious*, *cautious*, *vanilla*, *confident*, *very confident*

} are described as follows. Section B.1 provides the prompts for the CoT setting, while Section B.2 covers the prompts for the non-CoT setting.

**Discussion on Instruction-Following Ability.** Instruction-following is a prerequisite. Verbalized confidence assumes the model can return answer–confidence pairs in a fixed format. In practice, smaller or less capable models may fail at this step, which we treat as a formatting failure rather than a limitation of steering. Once a model reliably produces answer–confidence pairs, we find it also follows our steering prompts (e.g., *very cautious* vs. *very confident*) and shifts its confidence accordingly (see Figure 5 for how the model's confidence changes with different steering levels).

**Discussion on Additional Query Cost.** We deliberately query the model multiple times under a small set of steering prompts. In our black-box setting, we cannot read internal signals (e.g., logits), so these extra queries give us the missing evidence about how stable both the answers and the confidence scores are. This is a test-time scaling trade-off: we spend a few more forward passes to obtain much better-calibrated confidence and more reliable failure detection, without any training or access to internals. For complex, multi-turn reasoning, the same idea applies—investing a handful of additional queries typically yields more dependable confidence estimates and thus more trustworthy traces.

## B.1 With CoT

*very cautious*. Read the question, analyze step by step, provide your answer and your confidence in this answer. Note: (1) The confidence indicates how likely you think your answer will be true. (2) You are making important decisions, thus you should avoid giving a wrong answer with high confidence. (3) You should be very cautious, and tend to give low confidence on almost all of the answers. \nUse the following format to answer:\"'Explanation: [insert step-by-step analysis here]\nAnswer and Confidence (0-100): [ONLY the {ANSWER_TYPE}; not a complete sentence], [Your confidence level, please only include the numerical number in the range of 0-100]%"'\nOnly give me the reply according to this format, don't give me any other words.

*cautious*. Read the question, analyze step by step, provide your answer and your confidence in this answer. Note: (1) The confidence indicates how likely you think your answer will be true. (2) You are making important decisions, thus you should avoid giving a wrong answer with high confidence. \nUse the following format to answer:\"'Explanation: [insert step-by-step analysis here]\nAnswer and Confidence (0-100): [ONLY the {ANSWER_TYPE}; not a complete sentence], [Your confidence level, please only include the numerical number in the range of 0-100]%"'\nOnly give me the reply according to this format, don't give me any other words.

*vanilla*. Read the question, analyze step by step, provide your answer and your confidence in this answer. Note: The confidence indicates how likely you think your answer is true.\nUse the following format to answer:\n"'Explanation: [insert step-by-step analysis here]\nAnswer and Confidence (0-100): [ONLY the {ANSWER_TYPE}; not a complete sentence], [Your confidence level, please only include the numerical number in the range of 0-100]%\n"'\nOnly give me the reply according to this format, don't give me any other words.

*confident*. Read the question, analyze step by step, provide your answer and your confidence in this answer. Note: (1) The confidence indicates how likely you think your answer will be true. (2) You are making important decisions, thus you should avoid giving a right answer with low confidence. \nUse the following format to answer:\"'Explanation: [insert step-by-step analysis here]\nAnswer and Confidence (0-100): [ONLY the {ANSWER_TYPE}; not a complete sentence], [Your confidence level, please only include the numerical number in the range of 0-100]%"'\nOnly give me the reply according to this format, don't give me any other words.

*very confident*. Read the question, analyze step by step, provide your answer and your confidence in this answer. Note: (1) The confidence indicates how likely you think your answer will be true. (2) You are making important decisions, thus you should avoid giving a right answer with low confidence. (3) You should be very confident, and tend to give high confidence on almost all of the answers. \nUse the following format to answer:\"'Explanation: [insert step-by-step analysis here]\nAnswer and Confidence (0-100): [ONLY the {ANSWER_TYPE}; not a complete sentence], [Your confidence level, please only include the numerical number in the range of 0-100]%"'\nOnly give me the reply according to this format, don't give me any other words.

## B.2 Without CoT

*very cautious*. Read the question, provide your answer and your confidence in this answer. Note: (1) The confidence indicates how likely you think your answer will be true. (2) You are making important decisions, thus you should avoid giving a wrong answer with high confidence. (3) You should be very cautious, and tend to give low confidence on almost all of the answers. \nUse the following format to answer:\n"Answer and Confidence (0-100): [ONLY the ANSWER_TYPE; not a complete sentence], [Your confidence level, please only include the numerical number in the range of 0-100]%"\nOnly the answer and confidence, don't give me the explanation.

*cautious*. Read the question, provide your answer and your confidence in this answer. Note: (1) The confidence indicates how likely you think your answer will be true. (2) You are making important decisions, thus you should avoid giving a wrong answer with high confidence. \nUse the following format to answer:\n"Answer and Confidence (0-100): [ONLY the ANSWER_TYPE; not a complete sentence], [Your confidence level, please only include the numerical number in the range of 0-100]%"\nOnly the answer and confidence, don't give me the explanation.

*vanilla*. Read the question, provide your answer and your confidence in this answer. Note: The confidence indicates how likely you think your answer is true.\nUse the following format to answer:\n"Answer and Confidence (0-100): [ONLY the ANSWER_TYPE; not a complete sentence], [Your confidence level, please only include the numerical number in the range of 0-100]%"\nOnly the answer and confidence, don't give me the explanation.

*confident*. Read the question, provide your answer and your confidence in this answer. Note: (1) The confidence indicates how likely you think your answer will be true. (2) You are making important decisions, thus you should avoid giving a right answer with low confidence.\nUse the following format to answer:\n"Answer and Confidence (0-100): [ONLY the ANSWER_TYPE; not a complete sentence], [Your confidence level, please only include the numerical number in the range of 0-100]%"\nOnly the answer and confidence, don't give me the explanation.

*very confident*. Read the question, provide your answer and your confidence in this answer. Note: (1) The confidence indicates how likely you think your answer will be true. (2) You are making important decisions, thus you should avoid giving a right answer with low confidence. (3) You should be very confident, and tend to give high confidence on almost all of the answers. \nUse the following format to answer:\n"Answer and Confidence (0-100): [ONLY the ANSWER_TYPE; not a complete sentence], [Your confidence level, please only include the numerical number in the range of 0-100]%"\nOnly the answer and confidence, don't give me the explanation.

# C  Effects of Steering

To further understand the effects of steering on confidence, we present the confidence histograms of SteerConf with different steering levels from *very cautious* to *very confident* on LLaMA3 with CoT on DateUnd, GSM8K, Ethics, StrategyQA and Law datasets in Figure 5. Observe that the confidence distributions of SteerConf gradually change with varied steering levels. For example, on the StrategyQA dataset, the confidence distribution of *very cautious* is more conservative than that of *very confident*. This demonstrates the effectiveness of the proposed prompt steering strategy in SteerConf.

Table 5 demonstrates the effects of steering by presenting the differences in various metrics between the steered confidence distribution and the vanilla verbalized confidence distribution for GPT-3.5 without CoT. The steering effects are pronounced for high-intensity steering levels (*very cautious*, *very confident*) across most datasets, aligning with the intended directions. For instance, in the GSM8K dataset, the JS Div metric for *very cautious* is reduced by 52.11% compared to *vanilla*, while for *very confident*, it increases by 23.72%. However, for moderate-intensity levels such as *confident*, the steering direction may deviate from expectations. For example, in the Ethics dataset, the Mean metric for *confident* is 1.79 lower than *vanilla*. This highlights the importance of carefully designing the intensity of steering to achieve the desired effects.

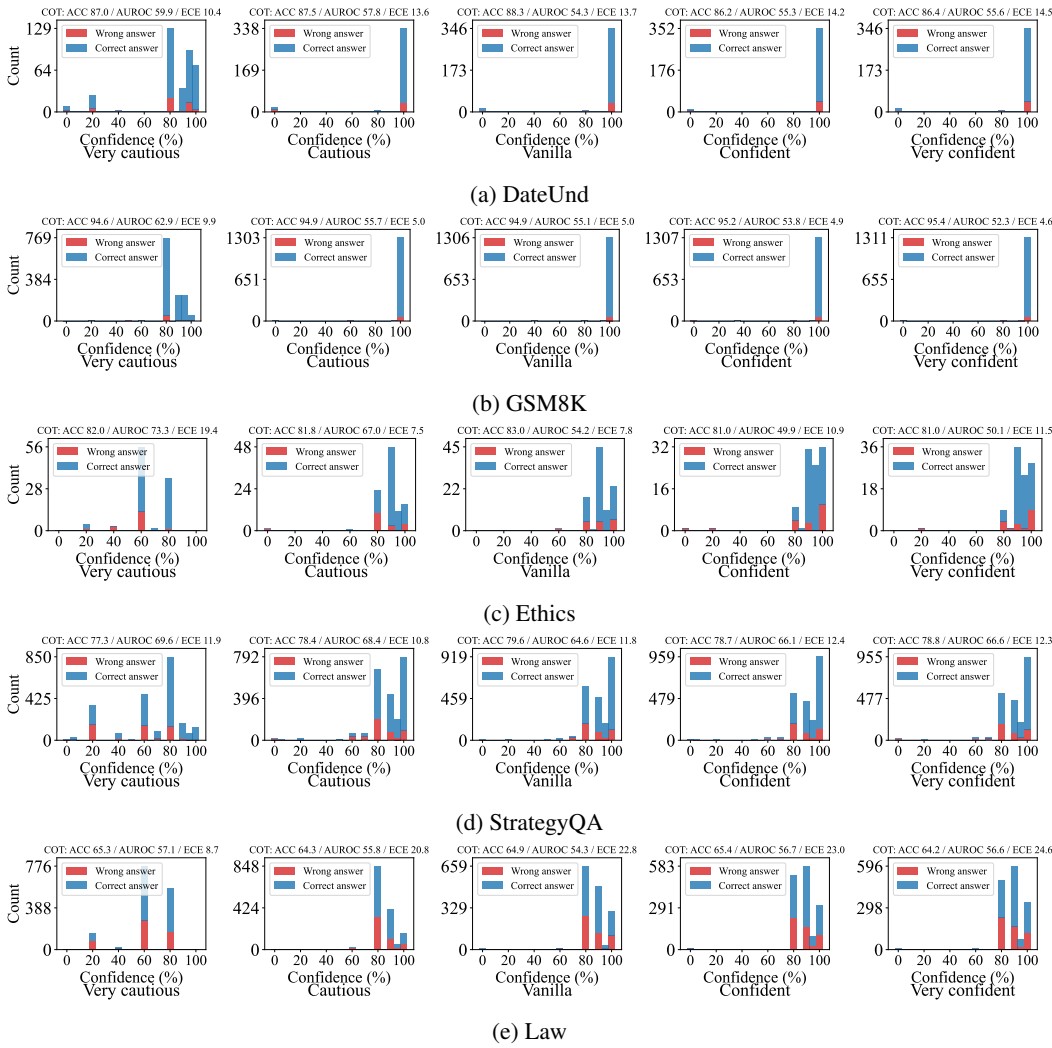

Figure 5: Confidence histograms for LLaMA3 with CoT on DateUnd, GSM8K, Ethics, StrategyQA and Law datasets. The columns are different steering levels.

# D Per-Prompt Ablation

To assess the contribution of each steering prompt, we perform a per-prompt ablation using LLaMA3 with CoT as the backbone model. For example, "SteerConf w/o *very confident* " removes the *very confident* prompt. Results are reported in Table 6. We observe that cautious prompts (*cautious*, *very cautious*) chiefly improve AUROC, whereas confident prompts (*confident*, *very confident*) primarily reduce ECE.

# E Results with a Smaller Model

We conduct experiments using the small model Qwen3-1.7b with the CoT setting, as shown in Table 8. Due to the model's relatively weaker instruction-following ability, frequent collapsed verbalized confidence elicitation renders results on some datasets (Law and Ethics) unusable. Therefore, we report results only on datasets with reliable responses. The results demonstrate that our proposed method consistently outperforms the vanilla verbalized confidence approach with the CoT setting, even with a small model backbone.

| Dataset | Was Dis | JS Div | Mean |
|---|---|---|---|
| Sport | -13.83 | -41.74 | -14.66 |
| Ethics | -21.20 | -72.34 | -20.49 |
| Law | -18.49 | -59.99 | -17.74 |
| ObjCnt | -11.42 | -34.23 | -16.93 |
| StrategyQA | -15.14 | -46.66 | -16.94 |
| DateUnd | -19.27 | -60.82 | -25.33 |
| GSM8K | -24.77 | -52.11 | -25.98 |

(a) *very cautious* v.s. *vanilla*

| Dataset | Was Dis | JS Div | Mean |
|---|---|---|---|
| Sport | -0.27 | -4.18 | -0.83 |
| Ethics | -1.40 | -10.63 | -0.04 |
| Law | -1.11 | -8.13 | -1.59 |
| ObjCnt | -0.01 | -1.86 | -0.13 |
| StrategyQA | -0.28 | -3.47 | -0.99 |
| DateUnd | -0.08 | -3.19 | -1.22 |
| GSM8K | -0.50 | -3.34 | -1.02 |

(b) *cautious* v.s. *vanilla*

| Dataset | Was Dis | JS Div | Mean |
|---|---|---|---|
| Sport | -0.53 | -4.93 | -0.47 |
| Ethics | -1.20 | -10.08 | -1.79 |
| Law | -0.79 | -6.45 | -1.33 |
| ObjCnt | -0.05 | -2.79 | -0.09 |
| StrategyQA | -0.46 | -5.52 | -1.02 |
| DateUnd | -0.19 | -4.84 | -1.44 |
| GSM8K | -0.85 | -4.41 | -1.58 |

(c) *confident* v.s. *vanilla*

| Dataset | Was Dis | JS Div | Mean |
|---|---|---|---|
| Sport | 1.11 | 17.43 | 1.71 |
| Ethics | 1.30 | 14.31 | 2.05 |
| Law | 1.73 | 19.22 | 3.26 |
| ObjCnt | 0.08 | 2.63 | 0.08 |
| StrategyQA | 0.97 | 13.16 | 1.41 |
| DateUnd | -0.19 | -4.75 | -0.69 |
| GSM8K | 4.08 | 23.72 | 5.36 |

(d) *very confident* v.s. *vanilla*

Table 5: Effects of Prompt Steering measured by different metrics on GPT-3.5. The confidence distribution is treated as a discrete distribution over the interval [0,100%], divided into bins of length 5. Each sub-table (a)-(d) shows the difference in distributions between a specific steering level (e.g., *very cautious*) and the vanilla verbalized confidence elicitation. "Mean" represents the average confidence for the dataset. "Was Dis" refers to the Wasserstein Distance, and "JS Div" denotes the Jensen-Shannon Divergence. To indicate the steering direction, the sign of the "Mean" metric is applied to these non-negative distribution metrics.

| Method | GSM8K | | Law | | DateUnd | | StrategyQA | | Ethics | | **Avg** | |
|---|---|---|---|---|---|---|---|---|---|---|---|---|
| | ECE | AUROC | ECE | AUROC | ECE | AUROC | ECE | AUROC | ECE | AUROC | ECE | AUROC |
| SteerConf w/o *very confident* | 7.2 | 80.9 | 5.1 | 64.3 | 11.1 | 70.9 | 5.8 | 74.3 | 17.9 | 80.4 | 9.4 | 74.1 |
| SteerConf w/o *confident* | 7.4 | 80.8 | 5.3 | 64.6 | 11.1 | 69.8 | 5.9 | 74.3 | 16.0 | 81.4 | 9.1 | 74.2 |
| SteerConf w/o *vanilla* | 6.8 | 79.3 | 4.4 | 64.2 | 11.4 | 68.5 | 5.8 | 74.4 | 12.9 | 78.7 | 8.3 | 73.0 |
| SteerConf w/o *cautious* | 6.9 | 80.7 | 4.7 | 63.8 | 9.3 | 68.3 | 5.3 | 74.2 | 12.9 | 77.2 | 7.8 | 72.8 |
| SteerConf w/o *very cautious* | 3.1 | 78.0 | 13.3 | 65.7 | 11.3 | 69.9 | 7.6 | 72.4 | 6.3 | 78.0 | 8.3 | 72.8 |
| SteerConf | 6.1 | 81.2 | 3.3 | 64.7 | 10.0 | 70.7 | 4.3 | 74.5 | 13.6 | 81.2 | **7.5** | **74.5** |

Table 6: Per-prompt ablations on LLaMA3 (CoT) by removing one steering prompt at a time. Lower ECE values and higher AUROC values indicate better performance.

## F  Additional Comparison with Avg-Conf Baselines

Table 9 compares SteerConf with more aggregation schemes using the LLaMA3 backbone with CoT setting. The results show that our proposed method SteerConf outperforms all baselines, including Self-Random and Misleading variants with Consistency, Pair-Rank, and Avg-Conf aggregation. For instance, SteerConf achieves ECE of 7.5%, notably better than Self-Random with Avg-Conf that is 14.3%.

## G  Prompt Robustness: SteerConf (brief) vs. Original SteerConf

We provide experiments to demonstrate the robustness of our method to prompt variations. We modified the expression of the steering prompts to be more concise while preserving the core steering intent. For example, we simplified the "*very cautious*" prompt as follows:

**Original SteerConf**: Read the question, provide your answer and your confidence in this answer. Note: (1) The confidence indicates how likely you think your answer will be true. (2) You are making important decisions, thus you should avoid giving a wrong answer with high confidence. (3) You should be very cautious, and tend to give low confidence on almost all of the answers.

| Method | GSM8K | | Law | | DateUnd | | StrategyQA | | Ethics | | Avg | |
|---|---|---|---|---|---|---|---|---|---|---|---|---|
| | ECE | AUROC | ECE | AUROC | ECE | AUROC | ECE | AUROC | ECE | AUROC | ECE | AUROC |
| GPT-3.5 | 62.6 | 55.8 | 43.2 | 51.7 | 60.2 | 56.6 | 29.6 | 53.3 | 26.0 | 54.8 | 44.3 | 54.4 |
| +SteerConf (brief) | 9.6 | 82.4 | 9.3 | 61.3 | 12.1 | 55.0 | 21.0 | 58.4 | 14.8 | 68.1 | 13.3 | 65.0 |
| +SteerConf | 22.8 | 82.9 | 24.0 | 60.6 | 33.0 | 60.5 | 14.9 | 58.6 | 10.8 | 71.0 | 21.1 | 66.7 |

Table 7: Prompt robustness on GPT-3.5: SteerConf (brief) vs. original steering prompts SteerConf. Lower ECE and higher AUROC are better.

| Method | GSM8K | | Sport | | DateUnd | | StrategyQA | | ObjCnt | | Avg | |
|---|---|---|---|---|---|---|---|---|---|---|---|---|
| | ECE | AUROC | ECE | AUROC | ECE | AUROC | ECE | AUROC | ECE | AUROC | ECE | AUROC |
| Qwen3-1.7b | 25.5 | 54.7 | 37.2 | 56.0 | 45.5 | 62.4 | 37.2 | 55.5 | 32.4 | 65.3 | 35.5 | 58.8 |
| Qwen3-1.7b + SteerConf | 8.3 | 80.4 | 40.1 | 49.8 | 46.5 | 62.3 | 39.2 | 51.1 | 22.0 | 74.5 | 31.2 | 63.6 |

Table 8: Results on Qwen3-1.7b with CoT prompting. Lower ECE and higher AUROC are better.

**SteerConf (brief)**: Read the question, provide your answer and your confidence in this answer. Note: (1) The confidence indicates how likely you think your answer will be true. (2) Be very cautious, tend to give very low confidence on every answer.

We compare the modified and original SteerConf prompts using GPT-3.5 as the backbone, as shown in Table 7. Notably, SteerConf (brief) achieves lower ECE than SteerConf, and both outperform GPT-3.5. These results demonstrate the robustness of our method to prompt variations.

# H  Per Steering Prompt Performance

In Table 10, we report results using only the *very cautious*, *cautious*, *confident*, and *very confident* prompts in the LLaMA3 with CoT setting, respectively. SteerConf consistently achieves the best overall performance in terms of average ECE and AUROC, though a single prompt could outperform it on some specific metrics. This result validates the effectiveness of our aggregation component on these single steering prompts.

# I  Effect of Steering Levels ($\ell = 5$ vs. $\ell = 3$)

To assess the trade-off between performance and query cost, we vary the number of steering levels from 5 to 3 to evaluate their impact, using LLaMA3 with CoT.

Table 11 reports results on LLaMA3 with CoT, where "no_verys" omits the most extreme prompts (*very cautious* and *very confident*) and "no_mild" omits the mild ones (*cautious* and *confident*).

# J  Discussion on Answer Selection and Effects on Accuracy

In this section, we analyze the impact of the answer selection mechanism in the SteerConf method. We denote SteerConf (Majority) as a variant of SteerConf, where the answer selection is replaced with a majority voting mechanism [38]. Specifically, in SteerConf (Majority), the final answer is determined by the most frequent response. In cases of ties, the answer with the highest mean steered confidence score among the tied responses is selected. The results in Table 12 demonstrate the calibration effectiveness of our approach. For instance, with LLaMA3, the average ECE for SteerConf is 7.3%, compared to 7.8% for SteerConf (Majority). Similarly, for GPT-3.5, the average PR-N metric for SteerConf is 62.0%, significantly outperforming SteerConf (Majority) at 53.9%. Table 12 also highlights the trade-off between calibration and accuracy. While SteerConf achieves superior calibration compared to vanilla verbalized confidence elicitation, it incurs a slight reduction in accuracy. Conversely, SteerConf (Majority) offers marginally better accuracy than SteerConf but at the expense of slightly weaker calibration. For example, with LLaMA3, the average accuracy of SteerConf is 82.3%, compared to 83.6% for SteerConf (Majority), while vanilla LLaMA3 achieves 83.1%. In summary, SteerConf is recommended for scenarios prioritizing calibration, while SteerConf (Majority) is better suited for applications where accuracy is more critical.

| Method | GSM8K | | Law | | DateUnd | | StrategyQA | | Ethics | | Avg | |
|---|---|---|---|---|---|---|---|---|---|---|---|---|
| | ECE | AUROC | ECE | AUROC | ECE | AUROC | ECE | AUROC | ECE | AUROC | ECE | AUROC |
| Vanilla | 5.0 | 55.1 | 22.8 | 54.3 | 13.7 | 54.3 | 11.8 | 64.6 | 7.8 | 54.2 | 12.2 | 56.5 |
| Misleading + Consistency | 4.4 | 83.8 | 18.9 | 59.7 | 18.4 | 67.8 | 11.7 | 65.3 | 14.0 | 75.0 | 13.5 | 70.3 |
| Self-Random + Consistency | 2.2 | 79.3 | 27.1 | 64.6 | 7.2 | 66.8 | 17.3 | 60.1 | 15.9 | 55.3 | 13.9 | 65.2 |
| Misleading + Avg-Conf | 3.3 | 83.5 | 18.8 | 62.4 | 15.6 | 70.4 | 11.1 | 67.8 | 14.9 | 75.2 | 12.8 | 71.9 |
| Self-Random + Avg-Conf | 2.3 | 79.3 | 27.1 | 64.4 | 8.8 | 68.3 | 17.2 | 59.9 | 16.3 | 55.0 | 14.3 | 65.4 |
| Top-K + Pair-Rank | 10.2 | 61.1 | 23.0 | 49.7 | 27.5 | 52.2 | 21.1 | 55.0 | 11.1 | 52.4 | 18.6 | 54.1 |
| Top-K + Avg-Conf | 56.4 | 60.6 | 15.1 | 55.8 | 40.7 | 55.9 | 36.4 | 52.0 | 5.2 | 67.5 | 30.7 | 58.4 |
| SteerConf (ours) | 6.1 | 81.2 | 3.3 | 64.7 | 10.0 | 70.7 | 4.3 | 74.5 | 13.6 | 81.2 | **7.5** | **74.5** |

Table 9: Comparison with additional aggregation schemes on LLaMA3 with CoT, including Avg-Conf and Pair-Rank/Top-K. Lower ECE and higher AUROC are better.

| Method | GSM8K | | Law | | DateUnd | | StrategyQA | | Ethics | | Avg | |
|---|---|---|---|---|---|---|---|---|---|---|---|---|
| | ECE | AUROC | ECE | AUROC | ECE | AUROC | ECE | AUROC | ECE | AUROC | ECE | AUROC |
| *very cautious* | 9.9 | 62.9 | 8.7 | 57.1 | 10.4 | 59.9 | 11.9 | 69.6 | 19.4 | 73.3 | 12.1 | 64.6 |
| *cautious* | 5.0 | 55.7 | 20.8 | 55.8 | 13.6 | 57.8 | 10.8 | 68.4 | 7.5 | 67.0 | 11.6 | 60.9 |
| *vanilla* | 5.0 | 55.1 | 22.8 | 54.3 | 13.7 | 54.3 | 11.8 | 64.6 | 7.8 | 54.2 | 12.2 | 56.5 |
| *confident* | 4.9 | 53.8 | 23.0 | 56.7 | 14.2 | 55.3 | 12.4 | 66.1 | 10.9 | 49.9 | 13.1 | 56.4 |
| *very confident* | 4.6 | 52.3 | 24.6 | 56.6 | 14.5 | 55.6 | 12.3 | 66.6 | 11.5 | 50.1 | 13.5 | 56.2 |
| SteerConf | 6.1 | 81.2 | 3.3 | 64.7 | 10.0 | 70.7 | 4.3 | 74.5 | 13.6 | 81.2 | **7.5** | **74.5** |

Table 10: Single steering prompt performance on LLaMA3 with CoT. Lower ECE and higher AUROC are better.

| Method | GSM8K | | Law | | DateUnd | | StrategyQA | | Ethics | | Avg | |
|---|---|---|---|---|---|---|---|---|---|---|---|---|
| | ECE | AUROC | ECE | AUROC | ECE | AUROC | ECE | AUROC | ECE | AUROC | ECE | AUROC |
| SteerConf (no_verys) ($\ell = 3$) | 3.0 | 76.8 | 14.7 | 64.4 | 11.2 | 69.4 | 7.5 | 71.6 | 8.0 | 72.8 | 8.9 | 71.0 |
| SteerConf (no_mild) ($\ell = 3$) | 8.4 | 78.8 | 6.8 | 62.8 | 10.4 | 67.4 | 7.2 | 73.7 | 16.3 | 77.7 | 9.8 | 72.1 |
| SteerConf ($\ell = 5$) | 6.1 | 81.2 | 3.3 | 64.7 | 10.0 | 70.7 | 4.3 | 74.5 | 13.6 | 81.2 | **7.5** | **74.5** |

Table 11: Comparison of steering levels on LLaMA3 (CoT): $\ell = 5$ vs. $\ell = 3$. Lower ECE and higher AUROC are better.

| | | GSM8K | DateUnd | ObjCnt | StrategyQA | Sport | Law | Ethics | Avg |
|---|---|---|---|---|---|---|---|---|---|
| Acc↑ | GPT3.5 | 78.3 | 71.5 | 58.1 | 68.4 | 63.3 | 46.7 | 67.7 | 64.9 |
| | +SteerConf | 69.0 | 72.1 | 56.9 | 65.0 | 58.9 | 47.7 | 60.0 | 61.4 |
| | +SteerConf (Majority) | 78.5 | 77.2 | 63.8 | 68.3 | 64.2 | 49.8 | 66.0 | 66.8 |
| | LLaMA3 | 94.9 | 88.3 | 91.3 | 79.6 | 79.5 | 64.9 | 83.0 | 83.1 |
| | +SteerConf | 94.6 | 87.0 | 91.3 | 77.4 | 78.4 | 65.3 | 82.0 | 82.3 |
| | +SteerConf (Majority) | 95.9 | 88.6 | 94.1 | 78.7 | 81.1 | 65.5 | 81.0 | 83.6 |
| ECE↓ | GPT3.5 | 20.3 | 30.8 | 41.8 | 26.0 | 20.5 | 44.3 | 24.8 | 29.8 |
| | +SteerConf | 5.4 | 6.7 | 19.7 | 14.5 | 10.8 | 16.0 | 15.1 | 12.6 |
| | +SteerConf (Majority) | 9.4 | 9.0 | 13.5 | 14.5 | 15.6 | 14.8 | 13.8 | 12.9 |
| | LLaMA3 | 5.0 | 13.7 | 8.7 | 11.8 | 7.7 | 22.8 | 7.8 | 11.1 |
| | +SteerConf | 6.1 | 10.0 | 2.7 | 4.3 | 11.0 | 3.3 | 13.6 | 7.3 |
| | +SteerConf (Majority) | 6.7 | 9.9 | 2.4 | 5.7 | 13.7 | 3.3 | 13.2 | 7.8 |
| AUROC↑ | GPT3.5 | 56.2 | 49.8 | 50.1 | 56.4 | 62.7 | 53.0 | 65.2 | 56.2 |
| | +SteerConf | 84.9 | 76.0 | 82.5 | 67.5 | 66.4 | 61.5 | 86.7 | 75.1 |
| | +SteerConf (Majority) | 84.2 | 72.9 | 78.1 | 64.0 | 64.8 | 60.5 | 80.1 | 72.1 |
| | LLaMA3 | 55.1 | 54.3 | 50.0 | 64.6 | 74.1 | 54.3 | 54.2 | 58.1 |
| | +SteerConf | 81.2 | 70.7 | 87.6 | 74.5 | 79.4 | 64.7 | 81.2 | 77.0 |
| | +SteerConf (Majority) | 75.1 | 67.3 | 82.4 | 72.9 | 75.4 | 64.2 | 82.8 | 74.3 |
| PR-N↑ | GPT3.5 | 26.5 | 28.0 | 42.0 | 36.8 | 46.9 | 55.5 | 45.7 | 40.2 |
| | +SteerConf | 72.3 | 49.7 | 74.7 | 48.9 | 52.2 | 60.9 | 75.0 | 62.0 |
| | +SteerConf (Majority) | 65.0 | 42.5 | 63.6 | 42.2 | 45.8 | 58.2 | 60.3 | 53.9 |
| | LLaMA3 | 12.4 | 13.4 | 8.7 | 29.7 | 38.3 | 38.2 | 26.1 | 23.8 |
| | +SteerConf | 41.3 | 24.0 | 49.0 | 43.2 | 47.5 | 50.6 | 51.7 | 43.9 |
| | +SteerConf (Majority) | 21.8 | 19.1 | 33.8 | 39.9 | 37.0 | 48.3 | 58.8 | 37.0 |
| PR-P↑ | GPT3.5 | 80.5 | 71.5 | 58.2 | 71.5 | 70.2 | 48.4 | 75.8 | 68.0 |
| | +SteerConf | 90.8 | 88.4 | 83.2 | 77.7 | 75.0 | 57.3 | 91.8 | 80.6 |
| | +SteerConf (Majority) | 93.0 | 89.3 | 83.4 | 78.0 | 76.9 | 58.1 | 90.3 | 81.3 |
| | LLaMA3 | 95.4 | 89.3 | 91.3 | 85.2 | 89.2 | 66.5 | 83.0 | 85.7 |
| | +SteerConf | 97.8 | 92.6 | 98.1 | 90.0 | 92.8 | 73.6 | 93.1 | 91.1 |
| | +SteerConf (Majority) | 97.8 | 92.8 | 98.1 | 90.0 | 92.7 | 73.5 | 93.2 | 91.2 |

Table 12: Performance comparison across datasets for SteerConf variants with different answer selection methods. Results include GPT-3.5 and LLaMA3, both under CoT settings.

