# OpenReview forum: "SteerConf: Steering LLMs for Confidence Elicitation"
_NeurIPS.cc/2025/Conference — NeurIPS 2025 poster_

### Official Review · Reviewer_BqFQ · 2025-06-30

**Clarity:** 2
**Significance:** 2
**Originality:** 2
**Rating:** 4
**Confidence:** 3

**Summary:**

This paper proposes SteerConf, a framework to improve confidence calibration in LLMs by steering their confidence scores via prompts. It introduces three components: (1) a steering prompt strategy; (2) a steered confidence consistency measure; and (3) a steered confidence calibration method. Without additional training or fine-tuning, SteerConf reduces calibration errors and improves failure detection across 7 benchmarks.

**Questions:**

1. How much impact does the setting of $l$ in the steering level have on the results, or as least how much $l$ is needed to achieve good results?
2. What does the argument in Line 186-187 mean, by saying "howerver overlooks the consistency of the verbalized confidences returned by LLMs, which may contain rich information of model's interior certainty"?
3. When using different prompts to guide the LLM's confidence elicitation, have the authors considered the influence of LLM's instruction following ability on these results?
4. The tested LLMs in this paper are all with large-scale parameters. What will the performance of LLMs with relatively less parameters (like 7B, 32B, etc.) be like? (As it is time-consuming to conduct experiments during the rebuttal, the authors can first discuss the relevant results.)

**Ethical Concerns:**

["NO or VERY MINOR ethics concerns only"]

**Final Justification:**

The authors' rebuttal has addressed most of my previous concerns. However, as the authors stated that "our primary focus is on widely-used LLMs with reasonably good capabilities for black-box confidence calibration (e.g., GPT-3.5, LLaMA3, GPT-4)", it implies that the SteerConf is a method relying on the strong reasoning capability of large language models, which limits it application to relatively small models (like 7B, 14B, etc.).

Overall, according to the novelty of the paper, the experimental results, and the authors' rebuttal, I think it is a boardline paper. So I rate it 4.

**Limitations:**

yes

**Paper Formatting Concerns:**

No formatting concerns on this paper.

**Quality:**

2

**Strengths And Weaknesses:**

**Strengths**

1. The idea of steering LLM confidence through carefully designed prompts is simple yet underexplored. And the SteerCof method does not require training or fine-tuning, making it applicable to black-box commercial LLMs.
2. This paper presents thorough experiments and evaluations across 7 diverse benchmarks (including commonsense, math, ethics, etc.) with multiple LLMs (GPT-3.5, GPT-4, LLaMA 3), demonstrating broad application of the framework.

**Weaknesses**

1. The proposed method requires multiple LLM calls per query, which is costly and makes it hard to extend to complex multi-turn reasoning tasks.
2. The steering prompts are manually designed. Small changes in these prompts might affect the final performance, but no robustness analysis is provided.
3. The answer selection process in Sec 3.3 is not intuitive. As claimed before that confidence scores elicited from LLMs are often overestimated, matching calibrated confidence $c(x)$ with steered confidence scores to select the final answer seems not quite reasonable.

---

> ### Author Rebuttal · Authors · 2025-07-31
>
> # Response to reviewer **BqFQ**
> We thank the reviewer for the insightful comments. We have addressed them with clarifications and experiments.
>
> **W1: multiple LLM calls per query; complex multi-turn reasoning tasks.**
>
> **Response:**
> We clarify that multiple query-based calibration is a common practice in the literature [31,37]. In our setting, since we don't have access to the internal states of LLMs, additional cost with multiple calls represents a necessary and reasonable trade-off to alleviate this information gap.
> For complex multi-turn reasoning tasks, our multi-query methodology can be viewed as a form of test-time scaling: investing more computational resources during inference to obtain better calibration and consequently more reliable reasoning traces.
>
> **W2: robustness of prompts**
>
> **Response:**
> We provide experiments to demonstrate the robustness of our method to prompt variations. We modified the expression of steering prompts to be more concise while preserving the core steering intent. For example, we simplified the "very_cautious" prompt as follows:
>
> *Original(SteerConf)*: Read the question, provide your answer and your confidence in this answer. Note: (1) The confidence indicates how likely you think your answer will be true. (2) You are making important decisions, thus you should avoid giving a wrong answer with high confidence. (3) You should be very cautious, and tend to give low confidence on almost all of the answers.
>
> *Modified(SteerConf_brief)*: Read the question, provide your answer and your confidence in this answer. Note: (1) The confidence indicates how likely you think your answer will be true. (2) Be very cautious, tend to give very low confidence on every answer.
>
> We compare the modified and original versions of SteerConf with backbone LLM GPT-3.5 without CoT setting in the table below. Notably, SteerConf_brief achieves even lower ECE than SteerConf, and both outperform GPT-3.5-vanilla. These results demonstrate the robustness of our proposed method to prompt variations.
>
>
> |Method|GSM8K ECE|GSM8K AUROC|Law ECE|Law AUROC|DateUnd ECE|DateUnd AUROC|StrategyQA ECE|StrategyQA AUROC|Ethics ECE|Ethics AUROC|Average ECE|Average AUROC|
> |-|-|-|-|-|-|-|-|-|-|-|-|-|
> | GPT-3.5-vanilla | 62.6 | 55.8 | 43.2 | 51.7 | 60.2 | 56.6 | 29.6 | 53.3 | 26.0 | 54.8 | 44.3 | 54.4 |
> | SteerConf_brief | 9.6 | 82.4 | 9.3 | 61.3 | 12.1 | 55.0 | 21.0 | 58.4 | 14.8 | 68.1 | 13.3 | 65.0 |
> | SteerConf | 22.8 | 82.9 | 24.0 | 60.6 | 33.0 | 60.5 | 14.9 | 58.6 | 10.8 | 71.0 | 21.1 | 66.7 |
>
>
> **W3: The answer selection process in Sec 3.3 is not intuitive**
>
> **Response:**
> Our answer selection mechanism is motivated by the principle that answers with higher calibrated confidence should be selected when the model is more certain, and vice versa. Specifically, we compare the *relative rank* of confidence scores among candidate answers, as formalized in equation (6). Experimental results in Table 5 demonstrate that this selection strategy improves calibration. We will clarify this point in the revised version.
>
> **Q1: setting of the steering level**
>
> **Response:**
> To investigate the impact of steering levels $l$, we vary $l$ from $5$ to $3$ using LLaMA3-70b-4bit with CoT, and report results in the table below. "SteerConf-no_verys" excludes very_cautious and very_confident prompts, while "SteerConf-no_mild" excludes cautious and confident prompts. The results show that $l=5$ yields better performance than $l=3$. Increasing $l$ may further improve results but incurs higher query cost and requires designing more fine-grained prompts, which is non-trivial. In practice, we find $l=5$ offers a good balance. We will add this discussion in the revised version.
>
> |Method|GSM8K ECE|GSM8K AUROC|Law ECE|Law AUROC|DateUnd ECE|DateUnd AUROC|StrategyQA ECE|StrategyQA AUROC|Ethics ECE|Ethics AUROC|Average ECE|Average AUROC|
> |-|-|-|-|-|-|-|-|-|-|-|-|-|
> | SteerConf-no_verys | 3.0 | 76.8 | 14.7 | 64.4 | 11.2 | 69.4 | 7.5 | 71.6 | 8.0 | 72.8 | 8.9 | 71.0 |
> | SteerConf-no_mild | 8.4 | 78.8 | 6.8 | 62.8 | 10.4 | 67.4 | 7.2 | 73.7 | 16.3 | 77.7 | 9.8 | 72.1 |
> | SteerConf | 6.1 | 81.2 | 3.3 | 64.7 | 10.0 | 70.7 | 4.3 | 74.5 | 13.6 | 81.2 | **7.5** | **74.5** |
>
>
>
> **Q2: Clarify the argument in Line 186-187**
>
> **Response:**
> We clarify that existing study [37] focuses on answer consistency, not confidence score consistency. We argue that a model's internal certainty is reflected in both answer and confidence consistency. Therefore, our method is designed to consider both aspects. Empirical results in the experiments section validate this approach.
>
> **Q3:  influence of LLM's instruction following ability**
>
> **Response:**
> Prior verbalized confidence works [31,37] did not address the impact of LLMs' instruction following ability. In our setting, the vanilla verbalized confidence method [31,37] already poses a challenge: if an LLM (e.g., small LLaMA-2 models) has poor instruction following ability, it may fail to produce answer-confidence pairs in the required format. Once an LLM can reliably perform verbalized confidence tasks, its instruction following ability is sufficient for our steering confidence instructions. Figure 4 demonstrates that our steering prompts are successfully followed by current LLM backbones. We will clarify this point in the revised version.
>
>
> **Q4: smaller models**
>
> **Response:**
> As suggested, we conduct experiments using the small model Qwen3-1.7b under the CoT setting, as shown in the table below. Due to the model's relatively weaker instruction following ability, frequent collapsed verbalized confidence elicitation renders results on some datasets (Law and Ethics) unusable; thus, we report results only on datasets with reliable responses. The results demonstrate that our proposed method consistently outperforms the vanilla verbalized confidence approach under the CoT setting, even with a small model backbone.
>
> |Method|GSM8K ECE|GSM8K AUROC|Sport ECE|Sport AUROC|DateUnd ECE|DateUnd AUROC|StrategyQA ECE|StrategyQA AUROC|ObjCnt ECE|ObjCnt AUROC|Average ECE|Average AUROC|
> |-|-|-|-|-|-|-|-|-|-|-|-|-|
> | Qwen3-1.7b Vanilla | 25.5 | 54.7 | 37.2 | 56.0 | 45.5 | 62.4 | 37.2 | 55.5 | 32.4 | 65.3 | 35.5 | 58.8 |
> | Qwen3-1.7b + SteerConf | 8.3 | 80.4 | 40.1 | 49.8 | 46.5 | 62.3 | 39.2 | 51.1 | 22.0 | 74.5 | 31.2 | 63.6 |

---

> > ### Comment · Reviewer_BqFQ · 2025-08-06
> > **Response to rebuttal**
> >
> > Thank you for your responses! I appreciate the additional experiments and efforts the authors made during the rebuttal, which have addressed most of my concerns. However, I still have some follow-up questions regarding the authors' responses:
> >
> > **Q2**: If I did not misunderstand, Table 5 in Appendix D shows the results of considering only answer consistency (SteerConf Majority) and considering both answer consistency and confidence score consistency (SteerConf). Based on the results, it seems that SteerConf is comparable to SteerConf (Majority) across datasets and metrics. So the authors may need to emphasize the role of confidence calibration in decision-making in the paper. Besides, Appendix D should be referred to in the paper.
> >
> > **Q4**: Although the average ECE and AUROC indicate SteerConf is better than Vanilla on small models, detailed results on multiple datasets seem to indicate that this improvement is mainly attributed to the improvement on GSM8K and ObjCnt, and its performance on other datasets is not significant. It may raise concerns about the effectiveness of SteerConf on smaller models.
> >
> > As a side note, the $l$ in your response to Q1 should be 1 for "SteerConf-no_verys" and "SteerConf-no_mild", since there are $2l+1$ levels as defined in the paper.

---

> > > ### Author Response · Authors · 2025-08-06
> > >
> > > We are pleased that our rebuttal has addressed most of your concerns. Thank you!
> > >
> > > For **Q2**, in Table 5 of Appendix D, both SteerConf(Majority) and SteerConf utilize the same calibrated confidence $c(x)$ from Eq. (4), but differ in answer selection: SteerConf(Majority) relies solely on answer consistency, while SteerConf employs the proposed answer selection mechanism (Lines 231–236) that is guided by calibrated confidence. Thus, Table 5 demonstrates the improvement of our answer selection mechanism. As suggested, we will emphasize the role of confidence calibration in decision-making and reference Appendix D in the paper.
> > >
> > > For **Q4**, as noted in your review, it is time-consuming to conduct the experiments during rebuttal, but we really wanted to provide relevant results for our discussion. Therefore, we ran experiments on a rather small model, Qwen3-1.7b, which, as noted previously, has weaker instruction-following ability and sometimes produces unusable results, leading to performance fluctuations. Nevertheless, as observed, our method achieves better average performance across datasets. In this work, our primary focus is on widely-used LLMs with reasonably good capabilities for black-box confidence calibration (e.g., GPT-3.5, LLaMA3, GPT-4), and our main results show that our method outperforms baselines. We agree that exploring confidence calibration for very small LLMs with unstable capabilities is a promising future direction, which may also help strengthen small LLMs. Besides, $l$ should be 1 and $2l+1$ is 3. We will incorporate and fix these into the revised paper.
> > >
> > > Thank you for your insightful discussion. Please let us know if you have further feedback.

---

> > > > ### Comment · Reviewer_BqFQ · 2025-08-07
> > > >
> > > > Thank you for your further explanation, which mostly resolves my issue with Q2 and Q4. I will raise my score accordingly.

---

> > > > > ### Author Response · Authors · 2025-08-07
> > > > >
> > > > > Thank you for raising your score. We appreciate your great efforts.

---

> ### Author Response · Authors · 2025-08-06
> **Kind reminder for rebuttal discussion**
>
> Dear Reviewer BqFQ,
>
> Thank you for your effort in reviewing our paper. We have addressed your comments by conducting **experiments on the robustness of prompts, varying different steering levels, and testing smaller model**, all of which further demonstrate the effectiveness of our designs, and provided additional clarifications on the design and setting in our work accordingly.
>
> Please have a look, and we look forward to your feedback. Thank you.
>
> Best,
> Authors of SteerConf

---

### Official Review · Reviewer_NUu4 · 2025-07-01

**Clarity:** 2
**Significance:** 3
**Originality:** 2
**Rating:** 5
**Confidence:** 3

**Summary:**

This paper introduces a method to enable large-language models to output a calibrated confidence score alongside answers to questions. The method implemented builds upon previous work in this area, where answer consistency has been used as a measure of confidence, however here the authors also introduce confidence consistency, based on the consistency of confidence scores, when models are steered towards different degrees of confidence. This provides 2 signals for confidence that together in concert may provide a more reliable measure of confidence. The authors demonstrate that the confidence scores output using their approach are better calibrated than other approaches. Additionally, by using their score to select answers, the authors demonstrate an improvement in performance. This work has important implications for high stakes domains, where confidence may be an important component of decision-making.

**Questions:**

1. Did the authors experiment with a greater/different number of steering levels in the prompting?
2. Did the authors consider weighting the confidence and answer consistency terms in their confidence measure to observe how this changed the results?

**Ethical Concerns:**

["NO or VERY MINOR ethics concerns only"]

**Final Justification:**

I have updated my score to reflect the author's response to the weaknesses highlighted in my original review.

Overall, I think this paper introduces a novel method for improving confidence scores in language-models which is an important issue deserving attention. The paper is not without its limitations, though the authors address these in the paper and in their response. In the future I hope to see more research build upon the findings presented in this paper to further advance the field of confidence elicitation in language-models.

**Limitations:**

This method only works in the simple closed-form question/answering setting. The authors could be more clear about this limitation, pointing out that this method would not work (or would require adaptation) for cases where a language-model is outputting a free-form longer response to a question.

**Paper Formatting Concerns:**

* Typo “mannually” on line 318

**Quality:**

3

**Strengths And Weaknesses:**

## Strengths
* The ability to generate reliable confidence scores with black-box language-models is an important area of research and the authors make a notable contribution to this field.
* The confidence score is well designed with a clear rationale and appears to offer an improvement compared to other scores used in past research.
* The score is relatively well validated, comparing their approach to several competing approaches, and demonstrating an improvement across multiple dimensions.

## Weaknesses
### Major
* The authors discuss the improvements in confidence calibration offered by their method however they only consider one metric for calibration (expected calibration error). It might be more informative to also use calibration curves to better understand the nature of the improvement in calibration. See [1] for a description of different types of calibration. I think figure 4 is less relevant for the main results and could be moved to the appendix to make space for more interrogation of calibration here.
* It’s not clear that these results would hold across many model-types. The results only compare 3 models, and 2 of these are both from OpenAI, so are likely trained in a similar manner. It would be useful to know whether this approach also works for smaller models, like Llama-8b models or other model types.
* Steerconf is only compared to the vanilla confidence level, however Figure 1 clearly shows that a cautious prompt gives better results. It would be good to compare this approach to more cautious prompts, as a more difficult baseline, to demonstrate that steerconf does offer an improvement to just using more cautious prompts.
### Minor
* In the introduction (line 66), the authors state “the very cautious prompt improves calibration, achieving a 29% increase in Area Under the Receiver Operating Curve” however AUROC does not measure calibration, so please amend this to be clear.
* The Related Work section needs to mention some of the work that informs the baselines they later use as part of their validation (such as [37])
* The introduction to the overall method (lines 138-147) could be rephrased to better justify the rationale to the approach. Lines 217-225 very convincingly and clearly explained why the measure was defined as it was and could be moved to the introduction to better introduce the approach.

[1] Van Calster B, Nieboer D, Vergouwe Y, De Cock B, Pencina MJ, Steyerberg EW. A calibration hierarchy for risk models was defined: from utopia to empirical data. J Clin Epidemiol. 2016 Jun;74:167-76. doi: 10.1016/j.jclinepi.2015.12.005. Epub 2016 Jan 6. PMID: 26772608.

---

> ### Author Rebuttal · Authors · 2025-07-31
>
> # Response to reviewer **NUu4**
> We thank the reviewer for the insightful comments. We have addressed all comments with experiments and clarifications.
>
> **W1: calibration curves**
>
> **Response:**
> We will add the calibration curves and move Figure 4 to the appendix as suggested by the reviewer. As external URLs to diagrams are not permitted in the rebuttal, these updates will be made in the revision.
>
> **W2: smaller and different model**
>
> **Response:**
> As suggested, we conduct experiments using the small model Qwen3-1.7b under the CoT setting, as shown in the table below. Due to its relatively weaker instruction following ability, frequent collapsed verbalized confidence elicitation renders results on some datasets (Law and Ethics) unusable; thus, we report results only on datasets with reliable responses. The results demonstrate that our proposed method consistently outperforms the vanilla verbalized confidence approach under the CoT setting, even with a small model backbone.
>
> |Method|GSM8K ECE|GSM8K AUROC|Sport ECE| Sport AUROC|DateUnd ECE|DateUnd AUROC|StrategyQA ECE|StrategyQA AUROC|ObjCnt ECE|ObjCnt AUROC|Average ECE|Average AUROC|
> |-|-|-|-|-|-|-|-|-|-|-|-|-|
> | Qwen3-1.7b Vanilla | 25.5 | 54.7 | 37.2 | 56.0 | 45.5 | 62.4 | 37.2 | 55.5 | 32.4 | 65.3 | 35.5 | 58.8 |
> | Qwen3-1.7b + SteerConf | 8.3 | 80.4 | 40.1 | 49.8 | 46.5 | 62.3 | 39.2 | 51.1 | 22.0 | 74.5 | 31.2 | 63.6 |
>
>
> **W3: compare this approach to more cautious prompts**
>
> **Response:**
> As suggested, we report results using only the very_cautious, cautious, confident, and very_confident prompts in the LLaMA3-70b-4bit with CoT setting, respectively. SteerConf consistently achieves the best overall performance in terms of average ECE and AUROC, though a single prompt could outperform on specific metric. We will include this comparison in the revised version as recommended by the reviewer.
>
> |Method|GSM8K ECE|GSM8K AUROC|Law ECE|Law AUROC|DateUnd ECE|DateUnd AUROC|StrategyQA ECE|StrategyQA AUROC|Ethics ECE|Ethics AUROC|Average ECE|Average AUROC|
> |-|-|-|-|-|-|-|-|-|-|-|-|-|
> | vanilla | 5.0 | 55.1 | 22.8 | 54.3 | 13.7 | 54.3 | 11.8 | 64.6 | 7.8 | 54.2 | 12.2 | 56.5 |
> | very_cautious | 9.9 | 62.9 | 8.7 | 57.1 | 10.4 | 59.9 | 11.9 | 69.6 | 19.4 | 73.3 | 12.1 | 64.6 |
> | cautious | 5.0 | 55.7 | 20.8 | 55.8 | 13.6 | 57.8 | 10.8 | 68.4 | 7.5 | 67.0 | 11.6 | 60.9 |
> | confident | 4.9 | 53.8 | 23.0 | 56.7 | 14.2 | 55.3 | 12.4 | 66.1 | 10.9 | 49.9 | 13.1 | 56.4 |
> | very_confident | 4.6 | 52.3 | 24.6 | 56.6 | 14.5 | 55.6 | 12.3 | 66.6 | 11.5 | 50.1 | 13.5 | 56.2 |
> | SteerConf | 6.1 | 81.2 | 3.3 | 64.7 | 10.0 | 70.7 | 4.3 | 74.5 | 13.6 | 81.2 | **7.5** | **74.5** |
>
>
>
> **Minor W4 & Minor W6:**
>
> **Response:**
> As suggested, we will rephrase the relevant sentence to improve presentation.
>
> **Minor W5: discuss a related work**
>
> **Response:**
> As suggested, we will discuss the suggested related work.
>
> **Q1: experiment with a greater/different number of steering levels**
>
> **Response:**
> As suggested, we vary the number of steering levels from $5$ to $3$ to assess their impact, using LLaMA3-70b-4bit with CoT. In the table, "SteerConf-no_verys" omits the very_cautious and very_confident prompts, while "SteerConf-no_mild" omits the cautious and confident prompts. The results show that $l=5$ yields better performance than $l=3$. In practice, $l=5$ offers a good balance, as increasing the number of levels incurs additional query cost. We will include this discussion in the revised version.
>
> |Method|GSM8K ECE|GSM8K AUROC|Law ECE|Law AUROC|DateUnd ECE|DateUnd AUROC|StrategyQA ECE|StrategyQA AUROC|Ethics ECE|Ethics AUROC|Average ECE|Average AUROC|
> |-|-|-|-|-|-|-|-|-|-|-|-|-|
> | SteerConf-no_verys | 3.0 | 76.8 | 14.7 | 64.4 | 11.2 | 69.4 | 7.5 | 71.6 | 8.0 | 72.8 | 8.9 | 71.0 |
> | SteerConf-no_mild | 8.4 | 78.8 | 6.8 | 62.8 | 10.4 | 67.4 | 7.2 | 73.7 | 16.3 | 77.7 | 9.8 | 72.1 |
> | SteerConf | 6.1 | 81.2 | 3.3 | 64.7 | 10.0 | 70.7 | 4.3 | 74.5 | 13.6 | 81.2 | **7.5** | **74.5** |
>
>
>
> **Q2: weighting the confidence and answer consistency terms**
>
> **Response:**
> We clarify that our approach *multiplies* the confidence and answer consistency terms, $c(x)=\mu_c\cdot \kappa_{ans}\cdot \kappa_{conf}$ as shown in Equation (4) below line 225. This multiplicative design avoids the need to manually balance the two terms, and leverages the transferability advantage of multiplication. Introducing a weight parameter would not directly assign a specific weight to either term.

---

> > ### Comment · Reviewer_NUu4 · 2025-08-05
> >
> > Thank you for response. I would like to acknowledge the time and effort spent producing additional experimental results. You have responded to each of my weaknesses in a satisfactory manner and I have upgraded my rating to 5 accordingly.

---

> > > ### Author Response · Authors · 2025-08-05
> > >
> > > We are pleased to know that we have satisfactorily addressed your comments. Thank you for your effort in reviewing our work and upgrading the rating.

---

### Official Review · Reviewer_1saN · 2025-07-02

**Clarity:** 3
**Significance:** 2
**Originality:** 2
**Rating:** 4
**Confidence:** 4

**Summary:**

This paper introduces SteerConf, a novel black-box framework for improving the confidence calibration of large language models via semantic steering. Instead of relying on internal logits or fine-tuning, SteerConf applies a set of carefully crafted prompts with varying confidence attitudes to elicit multiple predictions and verbalised confidence scores. The method computes a final calibrated confidence by aggregating the average confidence with both answer and confidence consistency metrics. Extensive experiments on seven benchmarks using GPT-3.5, GPT-4, and LLaMA3 demonstrate that SteerConf consistently improves Expected Calibration Error (ECE) and failure prediction metrics compared to vanilla verbalised confidence and other existing baselines.

**Questions:**

1.	Have the authors conducted any ablation studies to assess the individual impact of each steering prompt?
2.	How does the framework perform when applied to tasks with longer reasoning chains or in the CoT setting? Is there any further analysis in such cases?
3.	How does SteerConf perform when the prompts intentionally guide the model to express overly high or low confidence? Is the method robust to such suggestive or misleading prompt cues?

**Ethical Concerns:**

["NO or VERY MINOR ethics concerns only"]

**Final Justification:**

The authors carefully responded to my previous questions. Based on the explanation and more experimental results, I have decided to raise my score.

**Limitations:**

yes

**Quality:**

3

**Strengths And Weaknesses:**

Strengths:
1.	The framework works in a black-box setting without requiring internal access or model fine-tuning.
2.	The method is conceptually simple and easy to apply to any LLM with API access.
Weaknesses：
1.	Although the proposed confidence consistency metric captures numerical stability across prompts, it does not consider potential semantic differences in the generated answers. In tasks involving more open or ambiguous questions, different prompts may lead to answers with similar confidence scores but different meanings. This aspect is not sufficiently discussed in the paper.
2.	The method requires multiple LLM calls per input but does not analyse whether all prompts are necessary. Without understanding each prompt’s contribution, the method may be less efficient than necessary.

---

> ### Author Rebuttal · Authors · 2025-07-31
>
> # Response to reviewer 1saN
> We thank the reviewer for the insightful comments and we have addressed them as follows.
>
> **W1: semantic differences in the generated answers, more open or ambiguous questions**
>
> **Response:**
> We clarify that, following prior work, our current tasks focus on numerical answers (GSM8K) or answers with predefined options (Law), where ambiguity is minimal and answer consistency is both sufficient and effective. We agree that addressing semantic differences in generated answers is important for more open-ended tasks, and will include this discussion as future work in the revised version.
>
> **W2: Each prompt’s contribution**
> **Q1:  individual impact of each steering prompt**
>
> **Response:**
> As suggested, to assess the contribution of each steering prompt, we conduct an ablation study by removing each prompt individually using LLaMA3-70b-4bit with CoT as the backbone. For example, "SteerConf w/o very_confident" excludes the very_confident prompt. The results are shown in the table below. Notably, the cautious prompts (cautious, very_cautious) primarily improve AUROC, while the confident prompts (confident, very_confident) mainly enhance ECE.
>
>
> |Method|GSM8K ECE|GSM8K AUROC|Law ECE|Law AUROC|DateUnd ECE|DateUnd AUROC|StrategyQA ECE|StrategyQA AUROC|Ethics ECE|Ethics AUROC|Average ECE|Average AUROC|
> |-|-|-|-|-|-|-|-|-|-|-|-|-|
> | SteerConf w/o very_confident | 7.2 | 80.9 | 5.1 | 64.3 | 11.1 | 70.9 | 5.8 | 74.3 | 17.9 | 80.4 | 9.4 | 74.1 |
> | SteerConf w/o confident | 7.4 | 80.8 | 5.3 | 64.6 | 11.1 | 69.8 | 5.9 | 74.3 | 16.0 | 81.4 | 9.1 | 74.2 |
> | SteerConf w/o vanilla | 6.8 | 79.3 | 4.4 | 64.2 | 11.4 | 68.5 | 5.8 | 74.4 | 12.9 | 78.7 | 8.3 | 73.0 |
> | SteerConf w/o cautious | 6.9 | 80.7 | 4.7 | 63.8 | 9.3 | 68.3 | 5.3 | 74.2 | 12.9 | 77.2 | 7.8 | 72.8 |
> | SteerConf w/o very_cautious | 3.1 | 78.0 | 13.3 | 65.7 | 11.3 | 69.9 | 7.6 | 72.4 | 6.3 | 78.0 | 8.3 | 72.8 |
> | SteerConf | 6.1 | 81.2 | 3.3 | 64.7 | 10.0 | 70.7 | 4.3 | 74.5 | 13.6 | 81.2 | **7.5** | **74.5** |
>
> **Q2: Perform with CoT setting**
>
> **Response:**
> We highlight that Table 2 in our paper (Page 7, above line 260) presents results for the CoT setting across different model backbones. CoT prompting consistently enhances the performance of all LLMs. Importantly, SteerConf continues to outperform the vanilla verbalized confidence approach, confirming the effectiveness of our method regardless of CoT usage.
>
> **Q3: How does SteerConf perform when the prompts intentionally guide the model to express overly high or low confidence?**
>
> **Response:**
> The design principle of our method SteerConf is precisely to intentionally **steer the confidence** output of LLMs. SteerConf effectively handles cases when prompts intentionally guide the model to express overly high or low confidence. In fact, this intentional steering mechanism is the key design of our SteerConf (Steering Confidence) method as stated in Sections 1 and 3, where "steering" refers to: generating confidence scores in a specified direction (e.g., conservative or optimistic) by employing prompts with varying degrees of caution or confidence. Specifically, we apply steering prompts like "You should adopt a very cautious approach and assign low confidence to most of your answers" to guide the LLM to express deliberately calibrated confidence levels. The steered confidence scores and answers are then aggregated to obtain more reliable confidence estimates.
> We will emphasize this point in the revised version.

---

### Official Review · Reviewer_ZMK7 · 2025-07-03

**Clarity:** 3
**Significance:** 3
**Originality:** 2
**Rating:** 3
**Confidence:** 5

**Summary:**

This paper proposes a new approach to obtain verbalized confidence scores that are better calibrated. Specifically, the approach prompts the model through different prompts and obtains a list of paired answers and confidence scores. Then the authors propose a way to aggregate these pairs to obtain a final confidence score. This approach demonstrate better calibration performance than the vanilla verbalized confidence scores and several other baselines.

**Questions:**

NA

**Ethical Concerns:**

["NO or VERY MINOR ethics concerns only"]

**Final Justification:**

This paper is limited at novelty in general -- methodologically it is like a new way of aggregating different verbose confidence scores. Several similar works exist previously, and the originality is not significant enough. The original submission is not at good quality I would say, as in my review, it lacks some baselines and proper ablations for the design choices, thus I gave an initial rating of 2.

The authors added additional baselines and ablation results in the rebuttal which addressed my concerns on the experiments, but my concern on the originality of the paper persists. I think this paper is very borderline and I don't have clear preference, I raise my score to 3 because the initial version was not good, but the AC or the authors can interpret my score as 3.5.

**Limitations:**

yes

**Quality:**

3

**Strengths And Weaknesses:**

#### Strengths

1. This paper studies an important problem to calibrate LLMs’ confidence scores, which could be helpful to control hallucination and risks.
2. The proposed approach is training-free and can be easily adopted.


#### Weaknesses

1. This paper proposes a new way to aggregate confidence and answer pairs, and it should compare with stronger baselines such as Avg-Conf and Pair-Rank in [1] – which is necessary also because the evaluation datasets of the two works are similar.
2. The proposed approach has several components, and I think it should be made clear the effectiveness of each component. For example, (1) how necessary is it to use carefully designed steering prompts? Would just semantically similar prompts suffice? Or maybe we can just use one prompt but sample multiple different responses? (2) Controlling the prompts fixed, how effective of different aggregation approaches?

---

> ### Author Rebuttal · Authors · 2025-07-31
>
> # Response to reviewer **ZMK7**
> We have addressed all your comments below with experiments. Thank you for the constructive comments.
>
> **W1: compare with baselines such as Avg-Conf and Pair-Rank**
>
> **Response:**
> 1. As noted in Line 267, the "Top-K" baseline already incorporates the Pair-Rank aggregation, which is specifically tailored for the Top-K approach.
> 2. For Avg-Conf, we report results using the LLaMA3-70b-4bit backbone with CoT in the table below. The results show that our proposed method outperforms all baselines, including Self-Random and Misleading variants with Consistency, Pair-Rank, and Avg-Conf aggregation. For instance, SteerConf achieves ECE of 7.5, notably better than Self-Random with Avg-Conf that is 14.3.
>
> |Method|GSM8K ECE|GSM8K AUROC|Law ECE|Law AUROC|DateUnd ECE|DateUnd AUROC|StrategyQA ECE|StrategyQA AUROC|Ethics ECE|Ethics AUROC|Average ECE|Average AUROC|
> |-|-|-|-|-|-|-|-|-|-|-|-|-|
> | Vanilla | 5.0 | 55.1 | 22.8 | 54.3 | 13.7 | 54.3 | 11.8 | 64.6 | 7.8 | 54.2 | 12.2 | 56.5 |
> | Misleading+Consistency | 4.4 | 83.8 | 18.9 | 59.7 | 18.4 | 67.8 | 11.7 | 65.3 | 14.0 | 75.0 | 13.5 | 70.3 |
> | Self-Random+Consistency | 2.2 | 79.3 | 27.1 | 64.6 | 7.2 | 66.8 | 17.3 | 60.1 | 15.9 | 55.3 | 13.9 | 65.2 |
> | Misleading+Avg-Conf | 3.3 | 83.5 | 18.8 | 62.4 | 15.6 | 70.4 | 11.1 | 67.8 | 14.9 | 75.2 | 12.8 | 71.9 |
> | Self-Random+Avg-Conf | 2.3 | 79.3 | 27.1 | 64.4 | 8.8 | 68.3 | 17.2 | 59.9 | 16.3 | 55.0 | 14.3 | 65.4 |
> | Top-K+Pair-Rank | 10.2 | 61.1 | 23.0 | 49.7 | 27.5 | 52.2 | 21.1 | 55.0 | 11.1 | 52.4 | 18.6 | 54.1 |
> | Top-K+Avg-Conf | 56.4 | 60.6 | 15.1 | 55.8 | 40.7 | 55.9 | 36.4 | 52.0 | 5.2 | 67.5 | 30.7 | 58.4 |
> | SteerConf | 6.1 | 81.2 | 3.3| 64.7 | 10.0 | 70.7 | 4.3 | 74.5 | 13.6 | 81.2 | **7.5** | **74.5** |
>
> **W2: make clear the effectiveness of each component. (1)  steering prompts (2) different aggregation approaches**
>
> **Response:**
> We perform an ablation study to assess the impact of steering prompting and aggregation components, using LLaMA3-70b-4bit with CoT as the backbone. For (1), removing steering prompting yields "SteerConf w/o Steering Prompting". For (2), we replace our proposed aggregation with Consistency and Avg-Conf, denoted as "SteerConf+Consistency" and "SteerConf+Avg-Conf". Ablating both components results in the Self-Random approach with Consistency or Avg-Conf aggregation.
> The results are reported in the table below, which demonstrates that each component contributes to improved performance, supporting the effectiveness of our method’s design.
>
> |Method|GSM8K ECE|GSM8K AUROC|Law ECE|Law AUROC|DateUnd ECE|DateUnd AUROC|StrategyQA ECE|StrategyQA AUROC|Ethics ECE|Ethics AUROC|Average ECE|Average AUROC|
> |-|-|-|-|-|-|-|-|-|-|-|-|-|
> | SteerConf w/o Steering Prompting | 2.3 | 80.0 | 13.2 | 66.7 | 10.0 | 70.2 | 6.1 | 72.5 | 10.2 | 59.0 | 8.4 | 69.7 |
> |Self-Random+Consistency | 2.2 | 79.3 | 27.1 | 64.6 | 7.2 | 66.8 | 17.3 | 60.1 | 15.9 | 55.3 | 13.9 | 65.2 |
> | SteerConf+Consistency | 2.4 | 71.7 | 25.9 | 63.8 | 7.5 | 68.2 | 17.2 | 61.4 | 14.3 | 68.5 | 13.5 | 66.7 |
> | Self-Random+Avg-Conf | 2.3 | 79.3 | 27.1 | 64.4 | 8.8 | 68.3 | 17.2 | 59.9 | 16.3 | 55.0 | 14.3 | 65.4 |
> | SteerConf+Avg-Conf | 2.8 | 71.3 | 25.8 | 63.3 | 8.6 | 67.1 | 17.4 | 61.3 | 16.7 | 73.5 | 14.3 | 67.3 |
> | SteerConf | 6.1 | 81.2 | 3.3 | 64.7 | 10.0 | 70.7 | 4.3 | 74.5 | 13.6 | 81.2 | **7.5** | **74.5** |

---

> > ### Comment · Reviewer_ZMK7 · 2025-08-06
> >
> > I appreciate the authors's new results that address my major concerns on the empirical results. I am willing to raise my score. I think this paper is a borderline after addressing these concerns due to novelty in general.

---

> > > ### Author Response · Authors · 2025-08-06
> > >
> > > Thank you for raising your score. As mentioned in your review, our approach is training-free and can be easily adopted, and we propose a new way to aggregate confidence and answer pairs, and this approach demonstrate better calibration performance. These points mentioned by you should reflect our novel contributions in this work. We appreciate your effort and recognizing our contributions.
> > >
> > > Best,

---

> ### Author Response · Authors · 2025-08-06
> **Kind reminder for rebuttal discussion**
>
> Dear Reviewer ZMK7,
>
> You have raised two points that are constructive and easy to address. We have addressed them with experiments. Please have a look, and we look forward to your feedback. We thank you for your effort in reviewing our paper.
>
> **1. Comparison with Pair-Rank and Avg-Conf**.
> In the paper, we compared SteerConf to Top-K with Pair-Rank aggregation, and in this rebuttal, we have included additional results with Avg-Conf. Our SteerConf method outperforms these baselines, demonstrating its effectiveness.
>
> **2. Ablation Study**.
> We have conducted ablation studies to show the effectiveness of each component in our method.
>
> Best,
> Authors of SteerConf

---

### Note · Authors · 2025-08-12

We thank the reviewers for your insightful feedback, which contributed to a successful rebuttal and upgraded ratings.

Our key contributions are as follows (mostly quoting from the reviews):
- **Novel method on an important problem.** We propose SteerConf, a novel black-box framework to address the important problem of confidence calibration for LLMs (in the review summary or strengths from ZMK7, NUu4, 1saN), without internal access or fine-tuning, and can be easily adopted to LLMs (1saN, ZMK7, BqFQ).
- **Notable contribution in an underexplored field.** The idea of steering LLM confidence through carefully designed prompts is simple yet underexplored (BqFQ). The work makes a notable contribution to the field to generate reliable confidence with black-box LLMs (NUu4), which could be helpful to control hallucination and risks (ZMK7).
- **Well-designed techniques.** The calibrated confidence is well designed with a clear rationale and offers improvements over existing studies (NUu4). SteerConf applies a set of carefully crafted prompts with varying confidence attitudes to elicit verbalized confidences, and computes a final calibrated confidence by considering both answer and confidence consistency (1saN).
- **Consistent improvements in extensive experiments.** Experiments on seven benchmarks demonstrate that SteerConf consistently improves over existing methods across multiple dimensions (1saN, ZMK7, NUu4, BqFQ).

In the rebuttal, we have addressed the comments from the reviewers, summarized below. We will incorporate all suggested content into the paper.

- For NUu4, we conducted experiments comparing cautious prompts, varied the number of steering levels, ran small models, and provided further clarifications.

- For ZMK7, we conducted experiments comparing with Avg-conf and Pair-Rank, and performed ablation studies on steering prompts and aggregation approaches to validate our effectiveness.

- For BqFQ, we tested the robustness of prompts, varied steering levels, ran small models, and provided additional clarifications, to address the concerns from the reviewer.

- For 1saN with easy-to-address comments, we performed ablation studies on each steering prompt, clarified the CoT experiments have provided in Table 2 of the paper, and further explained our design principles. Although discussion was not possible, we believe all comments from 1saN are addressed.

We thank all reviewers and Area Chairs for your efforts.

Best,
Authors of SteerConf

---

### Decision · Program_Chairs · 2025-09-17

**Decision:**

Accept (poster)

**Comment:**

This paper introduces SteerConf, a novel, training-free framework for improving the confidence calibration of black-box Large Language Models. The core idea is to use a set of carefully crafted prompts to steer the model into generating answers with varying levels of verbalized confidence (e.g., cautious, confident). The final calibrated confidence is then derived by aggregating these outputs, considering both the consistency of the answers and the consistency of the confidence scores themselves. The authors demonstrate through extensive experiments on seven diverse benchmarks that SteerConf significantly outperforms existing methods in calibration and failure detection. The primary strengths, as highlighted by the reviewers, are that the paper addresses an important and underexplored problem (ZMK7, NUu4, BqFQ), the proposed method is simple, practical, and broadly applicable as it is training-free and works in a black-box setting (1saN, ZMK7), and the claims are supported by consistent and significant empirical improvements (1saN, ZMK7, NUu4, BqFQ). The main weaknesses raised were concerns about the method's novelty, with one reviewer finding it an incremental aggregation technique (ZMK7), its reliance on the instruction-following capabilities of large models which may limit its effectiveness on smaller models (BqFQ), and the increased computational cost due to multiple queries per input (BqFQ). The decision to accept is based on the paper's clear and practical contribution, validated by a strong set of experiments. The authors' rebuttal was exemplary, providing substantial new experiments that addressed nearly all reviewer concerns, leading three of the four reviewers to either confirm a high score or raise their initial score.